# FAST-DETECTGPT: EFFICIENT ZERO-SHOT DETECTION OF MACHINE-GENERATED TEXT VIA CONDITIONAL PROBABILITY CURVATURE

**Guangsheng Bao**
Zhejiang University
School of Engineering, Westlake University
baoguangsheng@westlake.edu.cn

**Yanbin Zhao**
School of Mathematics, Physics and Statistics,
Shanghai Polytechnic University
zhaoyb553@nenu.edu.cn

**Zhiyang Teng**
Nanyang Technological University
zhiyang.teng@ntu.edu.sg

**Linyi Yang, Yue Zhang**[*]
School of Engineering, Westlake University
Institute of Advanced Technology, Westlake Institute for Advanced Study
{yanglinyi,zhangyue}@westlake.edu.cn

## ABSTRACT

Large language models (LLMs) have shown the ability to produce fluent and cogent content, presenting both productivity opportunities and societal risks. To build trustworthy AI systems, it is imperative to distinguish between machine-generated and human-authored content. The leading zero-shot detector, Detect-GPT (Mitchell et al., 2023), showcases commendable performance but is marred by its intensive computational costs. In this paper, we introduce the concept of *conditional probability curvature* to elucidate discrepancies in word choices between LLMs and humans within a given context. Utilizing this curvature as a foundational metric, we present *Fast-DetectGPT* [1], an optimized zero-shot detector, which substitutes DetectGPT's perturbation step with a more efficient sampling step. Our evaluations on various datasets, source models, and test conditions indicate that Fast-DetectGPT not only surpasses DetectGPT by a relative around 75% in both the white-box and black-box settings but also accelerates the detection process by a factor of 340, as detailed in Table 1.

| Method | 5-Model Generations ↑ | ChatGPT/GPT-4 Generations ↑ | Speedup ↑ |
|---|---|---|---|
| DetectGPT | 0.9554 | 0.7225 | 1x |
| Fast-DetectGPT | **0.9887**
(**relative↑ 74.7%**) | **0.9338**
(**relative↑ 76.1%**) | **340x** |

Table 1: Detection accuracy (measured in AUROC) and computational speedup for machine-generated text detection. The *white-box setting* (directly using the source model) is applied to the methods detecting generations produced by five source models (5-model), whereas the *black-box setting* (utilizing surrogate models) targets ChatGPT and GPT-4 generations. Results are averaged from data in Table 2 for the 5-model generations and Table 3 for ChatGPT/GPT-4, where the 'relative↑' is calculated by $(new - old)/(1.0 - old)$, representing how much improvement has been made relative to the maximum possible improvement. Speedup assessments were conducted using the XSum news dataset, with computations on a Tesla A100 GPU.

---

[*]Corresponding author.
[1]The code and data are released at https://github.com/baoguangsheng/fast-detect-gpt.

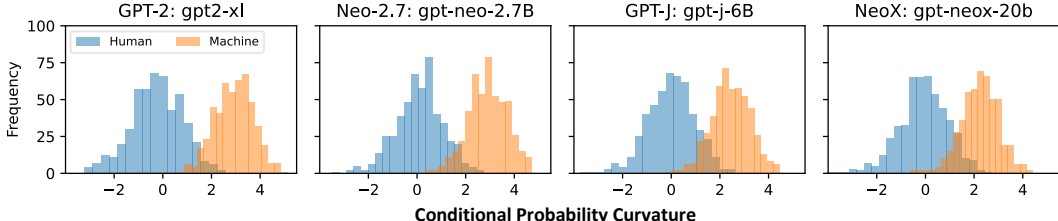

Figure 1: Distribution of *conditional probability curvatures* of the original human-written passages and the machine-generated passages by four source models on 30-token prefix from XSum.

# 1 INTRODUCTION

Large language models (LLMs) like ChatGPT (OpenAI, 2022), PaLM (Chowdhery et al., 2022), and GPT-4 (OpenAI, 2023) have dramatically influenced both industrial and academic landscapes. These models have transformed productivity in diverse fields such as news reporting, story writing, and academic research (M Alshater, 2022; Yuan et al., 2022; Christian, 2023). However, their misuse also introduces concerns—especially regarding fake news (Ahmed et al., 2021), malicious product reviews (Adelani et al., 2020), and plagiarism (Lee et al., 2023). The sheer fluency and coherence of content generated by these models make it challenging, even for experts, to determine its human or machine origin (Ippolito et al., 2020; Shahid et al., 2022). Addressing this issue necessitates reliable machine-generated text detection methods (Kaur et al., 2022; Chen & Shu, 2023).

Existing detectors can be grouped into two main categories: supervised classifiers (Solaiman et al., 2019; Fagni et al., 2021; Mitrović et al., 2023) and zero-shot classifiers (Gehrmann et al., 2019; Mitchell et al., 2023; Su et al., 2023). While supervised classifiers excel within their specific training domains, they falter when confronted with text from diverse domains or unfamiliar models (Bakhtin et al., 2019; Uchendu et al., 2020; Pu et al., 2023). Zero-shot classifiers, using a pre-trained language model directly without finetuning, are immune to domain-specific degradation and are on par with supervised classifiers on detection accuracy. This stems from their need for "universal features" that can function across multiple domains and languages (Gehrmann et al., 2019; Mitchell et al., 2023).

A typical zero-shot classifier, DetectGPT (Mitchell et al., 2023), works under the assumption that machine-generated text variations typically have lower model probability than the original, while human-written ones could go either way. Despite its effectiveness, employing probability curvature demands the execution of around one hundred model calls or interactions with services such as the OpenAI API to create the perturbation texts, leading to **prohibitive computational costs**.

In this paper, we posit a **new hypothesis** for detecting machine-generated text. By viewing text generation as a sequential decision-making process on tokens, our core assertion is that humans and machines exhibit discernible differences in token choice given a context. More specifically, machines lean towards tokens with higher statistical probability due to their pre-training on large-scale human-written corpus, while humans individually exhibit no such bias because they craft sentences based on underlying meanings, intentions, and contexts rather than data statistics. As a consequence, the conditional probability function $p(\tilde{x}|x)$ reaches its maximum point at a machine-generated $x$ (evidenced by a positive curvature at that point). Our empirical observation supports this hypothesis across diverse datasets and models, as Figure 1 illustrates. Specifically, the **conditional probability curvature** of machine-generated texts typically hovers around 3, whereas human-generated texts exhibit curvatures close to 0.

According to the above observation, we present **Fast-DetectGPT**, aiming to classify if a *passage* was produced by a particular *source model*, as outlined in Figure 2. In contrast to DetectGPT, our approach begins by sampling alternative word choices at each token (step 1). Subsequently, we assess the conditional probabilities of these generated samples (step 2) and combine them to arrive at a detection decision (step 3). Our empirical evaluation demonstrates the superior detection accuracy of Fast-DetectGPT over DetectGPT, showcasing a noteworthy relative boost of about 75% in both white-box and black-box settings. Intriguingly, in the black-box setting, Fast-DetectGPT even trumps DetectGPT's white-box performance by an average of 28%. Moreover, it aptly flags 80% of ChatGPT-crafted content, while only misidentifying 1% of human compositions.

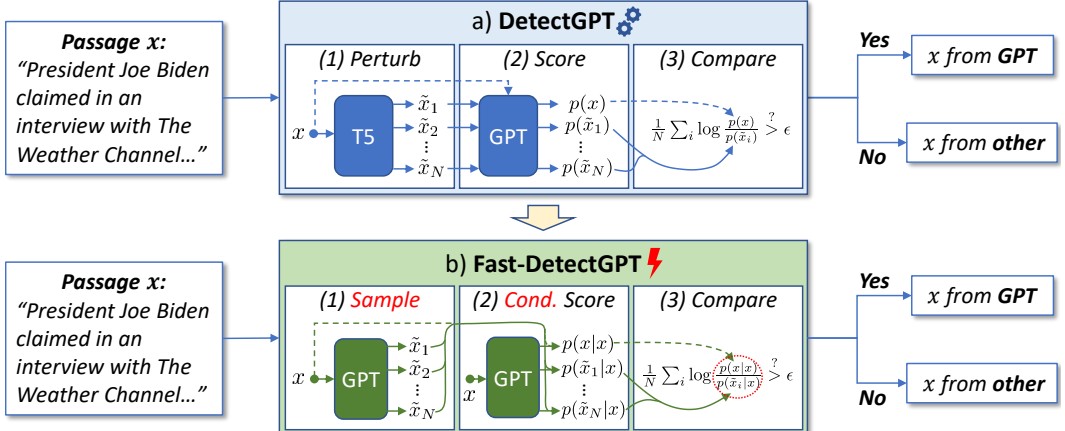

Figure 2: *Fast-DetectGPT* v.s. *DetectGPT* (Mitchell et al., 2023). Fast-DetectGPT uses a **conditional probability function** $p(\tilde{x}|x)$ as defined in Eq. 2. Notably, Fast-DetectGPT invokes the sampling GPT *once* to generate *all* samples and similarly calls the scoring GPT *once* to evaluate *all* samples, while DetectGPT interacts with the perturbation model T5 to produce *one* perturbation per call, and summons the scoring model GPT for each perturbation assessment. The threshold $\epsilon$ should be chosen to balance the false and true positive rates in practice.

Our main contributions are threefold: 1) unveiling and validation a new hypothesis that *human and machine select words differently given a context*, 2) proposing *conditional probability curvature* as a new feature to detect machine-generated text, *reducing the detection cost by two orders of magnitude*, and 3) achieving *the best average detection accuracy in both white-box and black-box settings* compare to existing zero-shot text detectors.

## 2 METHOD

### 2.1 TASK AND SETTINGS

Our objective is the zero-shot detection of machine-generated text, treating the challenge as a binary classification problem (detailed in Appendix A). Given a passage $x$, which may be human-authored or produced by a source model, the goal is to discern whether it is machine-generated.

In the **white-box setting**, we have the privilege of accessing the possible source model that a passage is either written by a human or generated by this source model. We use the source model to aid in scoring the candidate passage to inform the classification decision in the setting. Conversely, in the **black-box setting**, we operate without access to the source model. Instead, we rely on surrogate models to score the passage. Underpinning this approach is the assumption that language models, due to their training on vast human-authored corpora, inherently share characteristic features.

### 2.2 DETECTGPT BASELINE

Formally, given an input passage $x$ and the possible source model $p_\theta$, DetectGPT uses the source model for scoring (the white-box setting). Together with a predefined perturbation model $q_\varphi$, DetectGPT encapsulates the *probability curvature* as:

$$\mathbf{d}(x, p_\theta, q_\varphi) = \log p_\theta(x) - \mathbb{E}_{\tilde{x} \sim q_\varphi(\cdot|x)} \left[\log p_\theta(\tilde{x})\right], \qquad (1)$$

where $\tilde{x}$ is a perturbation produced by the masked language model $q_\varphi(\cdot|x)$. When $x$ emerges from sampling from the source model $p_\theta$, $\mathbf{d}(x, p_\theta, q_\varphi)$ tends to be positive, while for passage $x$ written by human, $\mathbf{d}(x, p_\theta, q_\varphi)$ tends to be zero. $p_\theta$ is also called the scoring model in this method, which is used to score the log probabilities.

**The Detection Process.** To estimate the expectation $\mathbb{E}_{\tilde{x} \sim q_\varphi(\cdot|x)} \log p_\theta(\tilde{x})$, DetectGPT employs a sampling approach. Typically, it generates around a hundred variations of the input text $x$ and then computes the average of the log probabilities associated with these variations. The detection process

is summarized as Figure 2a, where DetectGPT advocates a *three-step detection process*, which include: 1) Perturb – generating slight rewrites of the original text using a pre-trained mask language model; 2) Score – evaluating the probability of the text and its rewrites using a pre-trained GPT language model; 3) Compare – estimating the probability curvature and making the final decision accordingly.

**The Challenge.** The probability function $p_\theta(\tilde{x})$ models $\tilde{x}$ in a Markov chain. Even if disparities between $\tilde{x}$ and $x$ are slight, amounting to changes in merely about 15% of the tokens, the entire Markov chain demands reevaluation for accurate probability estimation. This slight variation within the Markov chain mandates invoking the scoring model afresh for each variation, as the Score step in Figure 2a denotes. In this paper, we deviate from assessing the probability function across the entire Markov chain. Instead, we focus on evaluating the conditional probability function for each individual token, thereby eliminating the need for repetitive scoring.

## 2.3 FAST-DETECTGPT

Fast-DetectGPT operates on the premise that humans and machines tend to select different words during the text-generation process, with machines exhibiting a propensity for choosing words with higher model probabilities. The hypothesis is rooted in the fact that LLMs, pre-trained on the large-scale corpus, mirror human collective writing behaviors instead of human individual writing behavior, resulting in a discrepancy in their word choices given a context.

The hypothesis is also substantiated to some extent by prior observations in the literature (Gehrmann et al., 2019; Hashimoto et al., 2019; Solaiman et al., 2019; Mitrović et al., 2023; Mitchell et al., 2023), which have indicated that machine-generated text typically boasts a higher average log probability (or lower perplexity) than human-written text. However, instead of solely relying on the assumption of a higher average log probability for machine-generated text, our approach posits the presence of a positive curvature within the conditional probability function specifically for machine-generated text.

Given a passage $x$ and a model $p_\theta$, we define the **conditional probability function** as

$$p_\theta(\tilde{x}|x) = \prod_j p_\theta(\tilde{x}_j|x_{<j}), \qquad (2)$$

where the tokens $\tilde{x}_j$ are independently predicted given $x$. As a special case, $p_\theta(x|x)$ equals to $p_\theta(x)$.

Specifically, we replace the probability function $p_\theta(\tilde{x})$ in DetectGPT with the conditional probability function $p_\theta(\tilde{x}|x)$. We estimate the curvature at the point $x$ by comparing the value of $p_\theta(x|x)$ with the values of alternative token choices $p_\theta(\tilde{x}|x)$. If $p_\theta(x|x)$ has a bigger value than $p_\theta(\tilde{x}|x)$, the function has a positive curvature at the point $x$, indicating that $x$ is more likely machine-generated. Otherwise, the function has a close-to-zero curvature at the point $x$, suggesting that $x$ is more likely human-written. We demonstrate the curvature distributions of human-written and machine-generated texts in Figure 1, where we can see that human-written texts are concentrated around the zero curvature.

Formally, given an input passage $x$ and the possible source model $p_\theta$ (the white-box setting), we quantify the **conditional probability curvature** as

$$\mathbf{d}(x, p_\theta, q_\varphi) = \frac{\log p_\theta(x|x) - \tilde{\mu}}{\tilde{\sigma}}, \qquad (3)$$

where

$$\tilde{\mu} = \mathbb{E}_{\tilde{x} \sim q_\varphi(\tilde{x}|x)} \left[ \log p_\theta(\tilde{x}|x) \right] \quad \text{and} \quad \tilde{\sigma}^2 = \mathbb{E}_{\tilde{x} \sim q_\varphi(\tilde{x}|x)} \left[ (\log p_\theta(\tilde{x}|x) - \tilde{\mu})^2 \right]. \qquad (4)$$

$\tilde{\mu}$ denotes the expected score of samples $\tilde{x}$ generated by the sampling model $q_\varphi(\cdot|x)$, and $\tilde{\sigma}^2$ the expected variance of the scores. We approximate $\tilde{\mu}$ using the average log probability of the random samples, and $\tilde{\sigma}^2$ using the mean of sample variances.

**Conditional Independent Sampling.** The independent sampling of alternative tokens is the key to the efficiency of Fast-DetectGPT. Specifically, we sample each token $\tilde{x}_j$ from $q_\varphi(\tilde{x}_j|x_{<j})$ given the fixed passage $x$ without depending on other sampled tokens. In practice, we can simply generate 10,000 samples (our default setting) by one line of PyTorch code: samples =

---

**Algorithm 1** Fast-DetectGPT machine-generated text detection.

**Input**: passage $x$, sampling model $q_\varphi$, scoring model $p_\theta$, and decision threshold $\epsilon$
**Output**: True – probably machine-generated, False – probably human-written.

1: **function** FASTDETECTGPT($x$, $q_\varphi$, $p_\theta$)
2:     $\tilde{x}_i \sim q_\varphi(\tilde{x}|x), i \in [1..N]$                 ▷ Conditional sampling
3:     $\tilde{\mu} \leftarrow \frac{1}{N}\sum_i \log p_\theta(\tilde{x}_i|x)$               ▷ Estimate the mean
4:     $\tilde{\sigma}^2 \leftarrow \frac{1}{N-1}\sum_i (\log p_\theta(\tilde{x}_i|x) - \tilde{\mu})^2$     ▷ Estimate the variance
5:     $\hat{\mathbf{d}}_x \leftarrow (\log p_\theta(x) - \tilde{\mu})/\tilde{\sigma}$     ▷ Estimate conditional probability curvature
6:     **return** $\hat{\mathbf{d}}_x > \epsilon$

---

torch.distributions.categorical.Categorical(logits=lprobs).sample([10000]), where the lprobs is the log probability distribution of $q_\varphi(\tilde{x}_j|x_{<j})$ for $j$ from 0 to the length of $x$.

The sampling process plays a pivotal role in guiding us toward the solution. To discern whether a token within a given context is machine-generated or human-authored, it is essential to compare it against a range of alternative tokens in the same context. By sampling a substantial number of alternatives (say 10,000), we can effectively map out the distribution of their $\log p_\theta(\tilde{x}_j|x_{<j})$ values. Placing the $\log p_\theta(x_j|x_{<j})$ value of the passage token within this distribution provides a clear view of its relative position, enabling us to ascertain whether it is an outlier or a more typical selection. This fundamental insight forms the core rationale behind the development of Fast-DetectGPT.

**The Detection Process.** As Figure 2b shows, Fast-DetectGPT proposes a new three-step detection process, including 1) *Sample* – we introduce a sampling model to generate alternative samples $\tilde{x}$ given the condition $x$, 2) *Conditional Score* – the conditional probability can be easily obtained by a single forward pass of the scoring model taking $x$ as the input. All the samples can be evaluated in the same predictive distribution, so we do not need multiple model calls, and 3) *Compare* – conditional probabilities of the passage and samples are compared to calculate the curvature. More implementation details are described in Algorithm 1.

We find that the "*Sample*" and "*Conditional Score*" steps can be merged and have an analytical solution instead of sampling approximation, as described in Appendix B. Furthermore, when we use the same model for sampling and scoring, the conditional probability curvature has a close connection to the simple Likelihood and Entropy baselines as follows.

**Connection to Likelihood and Entropy.** Utilizing a singular model for both sampling and scoring enables the combination of these processes into a single step, necessitating only one model call. Given this, the conditional probability curvature can be succinctly expressed as

$$\mathbf{d}(x, p_\theta) = \frac{\log p_\theta(x|x) - \tilde{\mu}}{\tilde{\sigma}}, \tag{5}$$

where $\tilde{\mu} = \mathbb{E}_{\tilde{x} \sim p_\theta(\tilde{x}|x)}[\log p_\theta(\tilde{x}|x)]$ and $\tilde{\sigma}^2 = \mathbb{E}_{\tilde{x} \sim p_\theta(\tilde{x}|x)}[(\log p_\theta(\tilde{x}|x) - \tilde{\mu})^2]$.

Intriguingly, the curvature's numerator reveals itself to be the sum of the baseline methods: Likelihood ($\log p_\theta(x)$) and Entropy ($-\tilde{\mu}$). While Likelihood and Entropy have been established as foundational baselines for zero-shot machine-generated text detection over the years (Lavergne et al., 2008; Gehrmann et al., 2019; Hashimoto et al., 2019; Mitchell et al., 2023), the discovery that their elementary combination can yield competitive detection accuracy was unforeseen.

## 3 EXPERIMENTS

### 3.1 SETTINGS

**Datasets.** We follow DetectGPT using six datasets to cover various domains and languages, including *XSum* for news articles (Narayan et al., 2018), *SQuAD* for Wikipedia contexts (Rajpurkar et al., 2016), *WritingPrompts* for story writing (Fan et al., 2018), *WMT16* English and German for different languages (Bojar et al., 2016), and *PubMedQA* for biomedical research question answering (Jin et al., 2019). We randomly sample 150 to 500 human-written examples per dataset as negative samples and produce equal numbers of positive samples by prompting the source model with the first 30 tokens of the human-written text (for PubMedQA, we only use the question as the prompt).

| Method | GPT-2 | OPT-2.7 | Neo-2.7 | GPT-J | NeoX | Avg. |
|---|---|---|---|---|---|---|
| **The White-Box Setting** | | | | | | |
| Likelihood | 0.9125 | 0.8963 | 0.8900 | 0.8480 | 0.7946 | 0.8683 |
| Entropy | 0.5174 | 0.4830 | 0.4898 | 0.5005 | 0.5333 | 0.5048 |
| LogRank | 0.9385 | 0.9223 | 0.9226 | 0.8818 | 0.8313 | 0.8993 |
| LRR | 0.9601 | 0.9401 | 0.9522 | 0.9179 | 0.8793 | 0.9299 |
| DNA-GPT ◇ | 0.9024 | 0.8797 | 0.869 | 0.8227 | 0.7826 | 0.8513 |
| NPR ◇ | 0.9948† | 0.9832† | 0.9883 | 0.9500 | 0.9065 | 0.9645 |
| DetectGPT (T5-3B/*) ◇ | 0.9917 | 0.9758 | 0.9797 | 0.9353 | 0.8943 | 0.9554 |
| Fast-DetectGPT (*/*) | **0.9967** | **0.9908** | 0.9940† | **0.9866** | **0.9754** | **0.9887** |
| *(Relative↑)* | *60.2%* | *62.0%* | *70.4%* | *79.3%* | *76.7%* | *74.7%* |
| **The Black-Box Setting** | | | | | | |
| DetectGPT (T5-3B/Neo-2.7) ◇ | 0.8517 | 0.8390 | 0.9797 | 0.8575 | 0.8400 | 0.8736 |
| Fast-DetectGPT (GPT-J/Neo-2.7) | 0.9834 | 0.9572 | **0.9984** | 0.9592† | 0.9404† | 0.9677† |
| *(Relative↑)* | *88.8%* | *73.4%* | *92.1%* | *71.4%* | *62.8%* | *74.5%* |

Table 2: Zero-shot detection on passages from *five source models*, averaging AUROCs across XSum, SQuAD, and WritingPrompts from detailed Table 5 in Appendix D.1. Typically, the source model is employed for scoring, except that DetectGPT (T5-3B/Neo-2.7) and Fast-DetectGPT (GPT-J/Neo-2.7) in a black-box setting utilize a surrogate Neo-2.7 model for scoring. While DetectGPT leverages T5-3B for perturbation generation, Fast-DetectGPT either resorts to the source model or a surrogate GPT-J for sample generation. † represents the second-best score. ◇ indicates methods that invoke models multiple times, thereby significantly increasing computational demands.

We evaluate the methods on machine-generated text produced by different source models from 1.3B to 1,800B, described in Appendix C.1. We call most of the models locally except GPT-3, ChatGPT, and GPT-4 through OpenAI API.

**Baselines.** For *zero-shot classifiers*, we mainly compare Fast-DetectGPT with *DetectGPT* (Mitchell et al., 2023), as well as its enhanced variant, *NPR* (Su et al., 2023) and *DNA-GPT* (Yang et al., 2023). These represent the baseline methodologies we aim to expedite. Furthermore, we juxtapose Fast-DetectGPT with established zero-shot techniques, such as *Likelihood* (mean log probabilities), *LogRank* (average log of ranks in descending order by probabilities), *Entropy* (mean token entropy of the predictive distribution) (Gehrmann et al., 2019; Solaiman et al., 2019; Ippolito et al., 2020), and LRR (an amalgamation of log probability and log-rank) (Su et al., 2023). Regarding *supervised classifiers*, Fast-DetectGPT is benchmarked against existing supervised classifiers, including GPT-2 detectors based on RoBERTa-base/large (Liu et al., 2019) crafted by OpenAI[2] and GPTZero (Tian & Cui, 2023).

## 3.2 MAIN RESULTS

We generate 500 samples per dataset among XSum, SQuAD, and WritingPrompts for the following experiments, measuring the detection accuracy in AUROC (see Appendix A).

**Inference Speedup.** We compare the inference time (excluding the time for initializing the model) of Fast-DetectGPT and DetectGPT on a Tesla A100 GPU using XSum generations from the five models in Table 2. Despite DetectGPT's use of GPU batch processing, splitting 100 perturbations into 10 batches, it still requires substantial computational resources. It totals 79,113 seconds (approximately 22 hours) over five runs. In contrast, Fast-DetectGPT completes the task in only 233 seconds (about 4 minutes), achieving a remarkable speedup factor of approximately 340x, highlighting its significant performance improvement.

**White-Box Zero-Shot Machine-Generated Text Detection.** We evaluate zero-shot methods using each source model for scoring and typically Fast-DetectGPT using the source model also for sampling. The averaged scores are shown in Table 2 with more detailed results per dataset reported in Table 5 in Appendix D.1. Fast-DetectGPT achieves the best average AUROC on the three datasets, outperforming DetectGPT by a relative 74.7% and its enhanced variant, NPR, by 68.2%. Notably, larger source models amplify this relative improvement, demonstrating the advantage of Fast-DetectGPT in detecting text produced by larger models.

---

[2]https://github.com/openai/gpt-2-output-dataset/tree/master/detector

| Method | ChatGPT | | | | GPT-4 | | | |
|---|---|---|---|---|---|---|---|---|
| | XSum | Writing | PubMed | Avg. | XSum | Writing | PubMed | Avg. |
| RoBERTa-base | 0.9150 | 0.7084 | 0.6188 | 0.7474 | 0.6778 | 0.5068 | 0.5309 | 0.5718 |
| RoBERTa-large | 0.8507 | 0.5480 | 0.6731 | 0.6906 | 0.6879 | 0.3821 | 0.6067 | 0.5589 |
| GPTZero | **0.9952** | 0.9292 | 0.8799 | 0.9348 | **0.9815** | 0.8262 | 0.8482 | 0.8853 |
| Likelihood (Neo-2.7) | 0.9578 | 0.9740 | 0.8775 | 0.9364 | 0.7980 | 0.8553 | 0.8104 | 0.8212 |
| Entropy (Neo-2.7) | 0.3305 | 0.1902 | 0.2767 | 0.2658 | 0.4360 | 0.3702 | 0.3295 | 0.3786 |
| LogRank(Neo-2.7) | 0.9582 | 0.9656 | 0.8687 | 0.9308 | 0.7975 | 0.8286 | 0.8003 | 0.8088 |
| LRR (Neo-2.7) | 0.9162 | 0.8958 | 0.7433 | 0.8518 | 0.7447 | 0.7028 | 0.6814 | 0.7096 |
| DNA-GPT (Neo-2.7) | 0.9124 | 0.9425 | 0.7959 | 0.8836 | 0.7347 | 0.8032 | 0.7565 | 0.7648 |
| NPR (T5-11B/Neo-2.7) | 0.7899 | 0.8924 | 0.6784 | 0.7869 | 0.5280 | 0.6122 | 0.6328 | 0.5910 |
| DetectGPT (T5-11B/Neo-2.7) | 0.8416 | 0.8811 | 0.7444 | 0.8223 | 0.5660 | 0.6217 | 0.6805 | 0.6228 |
| Fast-Detect (GPT-J/Neo-2.7) | 0.9907 | **0.9916** | **0.9021** | **0.9615** | 0.9067 | **0.9612** | **0.8503** | **0.9061** |
| *(Relative ↑)* | *94.1%* | *92.9%* | *61.7%* | *78.3%* | *78.5%* | *89.7%* | *53.1%* | *75.1%* |

Table 3: Detection of *ChatGPT* and *GPT-4* generations. The black-box settings are used for all zero-shot methods, where the Likelihood provides the strongest baseline. A comparison of GPT-3 generation detection is provided in Appendix D.2, where we observe a similar hierarchy in accuracy.

**Black-Box Zero-Shot Machine-Generated Text Detection.** Table 2 further contrasts Fast-DetectGPT and DetectGPT in a black-box setting, employing a surrogate model, Neo-2.7 (empirically superior among GPT-2, Neo-2.7, and GPT-J) for scoring. Fast-DetectGPT (GPT-J/Neo-2.7) achieves a relative AUROC enhancement of 74.5% over DetectGPT (T5-3B/Neo-2.7) across the datasets. Specifically, the boost in Wikipedia context (SQuAD) averages at 0.1682 AUROC (detailed in Table 5 in Appendix D.1). Moreover, Fast-DetectGPT (GPT-J/Neo-2.7) outperforms DetectGPT (T5-3B/*) by relatively 27.6% on average. Such outcomes solidify Fast-DetectGPT's potential in the black-box setting as a versatile text detector across diverse domains and source models.

**Ablation Study.** We study the impact of the choice of the sampling model $q_\varphi$ and the impact of the normalization $\tilde{\sigma}$ in Eq. 3 on the detection accuracy. Experiments show that a properly selected sampling model can further enhance the performance of Fast-DetectGPT in the white-box setting by relatively about 27%, while the normalization enhances the performance by 10%. More details are described in Appendix E.

## 3.3 RESULTS IN REAL-WORLD SCENARIOS

We further assess Fast-DetectGPT in a black-box setting using passages generated by GPT-3, Chat-GPT, and GPT-4 to simulate real-world scenarios. Using parameters and prompts delineated in Appendix C.2, we produced 150 samples per dataset and source model. As evidenced in Table 3, Fast-DetectGPT consistently exhibits superior detection proficiency. It surpasses DetectGPT by relative AUROC margins of 78.3% for ChatGPT and 75.1% for GPT-4. When compared to the supervised detectors RoBERTa-base/large and GPTZero, Fast-DetectGPT achieves overall higher accuracy. Collectively, these outcomes underscore the potential of Fast-DetectGPT working in real-world scenarios.

Interestingly, the commercial model GPTZero performs the best on news (XSum) but worse on stories (WritingPrompts) and technical writings (PubMedQA), indicating that the model may mainly be trained on news generations. The Likelihood detector emerges as the strongest baseline, which diverges from the results on open source models presented in Table 2, where Likelihood trails DetectGPT and NPR. A consistent trend is evident with GPT-3 generations (Appendix D.2). In comparison, Fast-DetectGPT offers more robust and consistent performance.

## 3.4 USABILITY ANALYSIS

**Interpretation of AUROC.** In real-world scenarios, our concern extends beyond overall detection accuracy; we are particularly interested in the recall (the true positive rate) while minimizing the likelihood of making type-I errors (achieving a low false positive rate). As depicted in Figure 3, when applied to ChatGPT-generated content, Fast-DetectGPT achieves a recall of 87% for machine-generated texts with only 1% misclassification of human-written text as machine-generated. When the tolerance for false positives increases to 10%, the recall reaches 98%. When applied to GPT-4

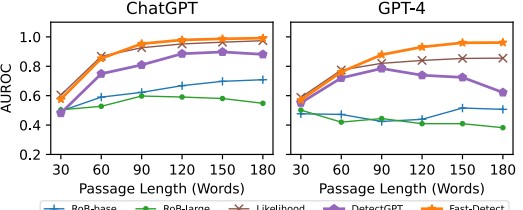

Figure 3: ROC curves in log scale evaluated on stories (WritingPrompts), where the dash lines denote the random classifier.

Figure 4: Detection accuracy (AUROC) on story passages (WritingPrompts) truncated to target number of words.

generations, the task becomes significantly more challenging but Fast-DetectGPT still achieves a recall of 89% on the condition of a false positive rate less than 10%. These outcomes underscore the potential of Fast-DetectGPT in effectively detecting texts generated by state-of-the-art large language models.

**Robustness on Different Passage Lengths.** Zero-shot detectors are supposed to perform worse on shorter passages due to their statistical nature. We conduct evaluations by truncating the passages to various target lengths. As Figure 4 illustrates, the detectors show consistent trends on passages produced by ChatGPT, where the detection accuracy generally increases for longer passages. In contrast, on passages from GPT-4, the detectors show inconsistent trends. Specifically, the supervised detectors show a performance decline when the passage length increases, while DetectGPT experiences an increase at the beginning followed by an unexpected decrease when the passage length exceeds 90 words. We speculate the non-monotonic trends of the supervised detectors and Detect-GPT are rooted in their perspective on handling the passages as a whole chain of tokens, which does not generalize to different lengths. In contrast, Fast-DetectGPT exhibits a consistent, monotonic increase in accuracy as passage length grows, indicating the robust performance of Fast-DetectGPT across passages of varying lengths.

**Robustness across Domains and Languages.** Detectors are expected to generalize to different domains and languages for higher usability. We compare Fast-DetectGPT against supervised detectors on both in-distribution and out-distribution datasets. Figure 5 reveals that Fast-DetectGPT achieves competitive detection accuracy on the in-distribution datasets XSum and WMT16-English. However, it significantly outperforms supervised detectors on the out-distribution datasets PubMedQA and WMT16-German. Moreover, it is noteworthy that Fast-DetectGPT consistently outperforms DetectGPT across all four datasets. These results underscore the robustness of Fast-DetectGPT across diverse domains and languages.

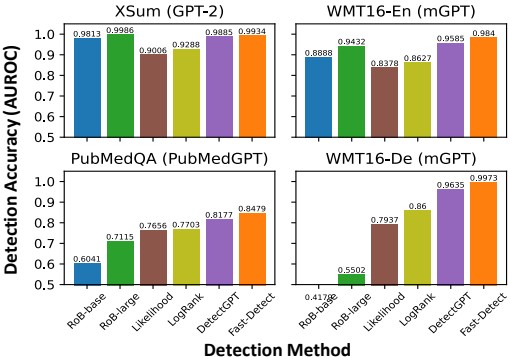

Figure 5: Compare with supervised detectors, evaluated in AUROC. We generate 200 test samples for each dataset and source model.

**Robustness against Decoding Strategies.** Machine systems employ various decoding strategies, including top-$k$ sampling, top-$p$ (Nucleus) sampling, and temperature sampling with a constant $T$. Our experiments evaluate these strategies using the five models and three datasets mentioned in Table 2 by setting $k = 40$, $p = 0.96$, and $T = 0.8$ for all cases. In the white-box setting, Fast-DetectGPT consistently emerged superior across all sampling strategies. It surpassed DetectGPT with relative margins of 95% in Top-$p$ sampling, 81% in Top-$k$ sampling, and a striking 99% in temperature sampling, as elaborated in Table 9 in Appendix F.1. Similar relative improvements are also achieved in the black-box setting. These results underscore the consistent performance of Fast-DetectGPT in detecting text generated through diverse decoding strategies.

**Robustness under Paraphrasing Attack.** We analyze the robustness under *paraphrasing attack* and propose *decoherence attack*, finding that Fast-DetectGPT consistently outperforms both zero-shot and trained detectors, as illustrated in Appendix F.2.

## 4 Discussion

Fast-DetectGPT performs about 65% better in white-box settings than black-box ones. Industrial services could leverage the white-box setting to enhance the content authorship for proprietary LLMs like ChatGPT and GPT-4 by exposing conditional probability curvature in the service API, without significant extra cost on the computation of the feature.

From the research perspective, the black-box setting may have unforeseen potential. The best model for this setting remains unclear, and may depend on factors like model size, training corpus breadth, and training process convergence. These factors warrant further investigation to provide clarity and guidance in the development of black-box detection methods.

We further discuss the broader implications of Fast-DetectGPT in Appendix G, covering *text authorship* and *watermarking*.

**Limitations and Future Work.** Our initial research impetus centered on accelerating the detection process of DetectGPT. However, Fast-DetectGPT not only accelerates this process but also brings about notable enhancements in detection accuracy. In this paper, our focus is predominantly on these empirical advancements, with theoretical explorations earmarked for future endeavors.

Furthermore, Fast-DetectGPT's design leans on pre-trained models to span a multitude of domains and languages. This presents a challenge in a black-box setting, as no single model can seamlessly span all linguistic territories and domains. This is due to the intrinsic nature of pre-trained models being tailored to specific domains and languages.

## 5 Related Work

Current research primarily concentrates on **supervised methods**, involving the training of classifiers to distinguish between machine-generated and human-written text (Jawahar et al., 2020). These classifiers are typically trained using bag-of-words (Solaiman et al., 2019; Fagni et al., 2021) or neural representations (Bakhtin et al., 2019; Solaiman et al., 2019; Uchendu et al., 2020; Ippolito et al., 2020; Fagni et al., 2021; Yan et al., 2023; Pu et al., 2023; Mitrović et al., 2023; Li et al., 2023). It has been observed that these trained classifiers often exhibit overfitting tendencies, adapting too closely to the specific distribution of text domains and source models during training (Bakhtin et al., 2019; Uchendu et al., 2020), which consequently leads to limited generalization capabilities when exposed to out-of-distribution data (Pu et al., 2023). To address this challenge, our research focuses on zero-shot detection, aiming to identify "universal features" that can be applied to different domains, source models, and languages.

Existing **zero-shot detectors** primarily rely on statistical features, leveraging pre-trained large language models to gather them. These features encompass a range of measures, including relative entropy and perplexity (Lavergne et al., 2008), bag-of-words, average probability, and top-K buckets (Gehrmann et al., 2019), likelihood (Hashimoto et al., 2019; Solaiman et al., 2019), probability curvature (DetectGPT) (Mitchell et al., 2023), normalized log-rank perturbation (NPR) (Su et al., 2023), and divergence between multiple completions of a truncated passage (DNA-GPT) (Yang et al., 2023). Due to their statistical nature, zero-shot methods generally exhibit higher detection accuracy on longer passages (Lavergne et al., 2008). In this paper, we introduce a novel "conditional probability curvature" for machine-generated text detection. Differing from previous probability curvature or completion divergence approaches that require numerous model calls (variating from 10 to 100), our new feature only necessitates a single model forward pass, significantly expediting the detection process. Importantly, this new feature markedly enhances detection accuracy.

## 6 Conclusion

Our investigation reveals that conditional probability curvature on the token level serves as a more fundamental indicator of machine-generated texts, validating our proposed hypothesis concerning the distinction between machine and human text generation processes. Building upon this new hypothesis, our detector Fast-DetectGPT accelerates upon DetectGPT by two orders of magnitude. Experimental results further demonstrate that Fast-DetectGPT also significantly improves detection accuracy by approximately 75% in both white-box and black-box settings.

ACKNOWLEDGMENTS

We would like to thank the anonymous reviewers for their valuable feedback. This work is funded by the National Natural Science Foundation of China Key Program (Grant No. 62336006) and the Pioneer and "Leading Goose" R&D Program of Zhejiang (Grant No. 2022SDXHDX0003). Yanbin Zhao is supported by the National Natural Science Foundation of China (Grant No. 12201158).

ETHICAL CONSIDERATIONS AND BROADER IMPACT

Fast-DetectGPT, serving as a highly efficient detector for machine-generated text, holds promise in enhancing the integrity of AI systems by combating issues like fake news, disinformation, and academic plagiarism. However, akin to other methods reliant on Large Language Models (LLMs), it is susceptible to inheriting biases present in the training data. Notably, as emphasized by Liang et al. (2023), LLM-based detection systems may exhibit an elevated false-positive rate when confronted with text from non-native English speakers. Given the widespread and diverse utilization of such technologies, this presents a notable concern.

An immediate suggestion is to substitute the underlying LLMs in Fast-DetectGPT with alternative models trained on more varied and representative corpora. Additionally, we advocate for community involvement in the ongoing efforts to develop more inclusive LLMs, a development that would benefit not only Fast-DetectGPT but also similar systems at large.

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

## A  Zero-Shot Detection Task and Settings

Our research centers on zero-shot detection of machine-generated text, under the premise that our model has not undergone training using any machine-generated text. This distinguishes our approach from conventional supervised methods, which commonly employ discriminative training strategies to acquire specific syntactic or semantic attributes customized for machine-generated text. In contrast, our zero-shot methodology capitalizes on the inherent capabilities of large language models to identify anomalies that function as markers of machine-generated content.

**The White-box Setting.** Conventional zero-shot methodologies often operate under the assumption that the source model responsible for generating machine-generated text is accessible. We refer to this context as the *white-box setting*, where the primary goal is to distinguish machine-generated texts produced by the source model from those generated by humans. In this white-box setting, our detection decisions are dependent on the source model, but it is not mandatory to possess detailed knowledge of the source model's architecture and parameters. For instance, within the white-box framework, a system like DetectGPT utilizes the OpenAI API to identify text generated by GPT-3, without requiring extensive knowledge of the inner workings of GPT-3.

**The Black-box Setting.** In real-world situations, there could be instances where we lack knowledge about the specific source models employed for content generation. This necessitates the development of a versatile detector capable of identifying texts generated by a variety of automated systems. We term this scenario the *black-box setting*, where the objective is to differentiate between machine-generated texts produced by diverse, unidentified models and those composed by humans. In this context, the term "black box" signifies that we lack access to information about the source model or any details pertaining to it.

**Evaluation Metric (AUROC).** Instead of measuring the detection accuracy with a specific threshold ($\epsilon$ in Figure 2), we measure the detection accuracy in the area under the receiver operating characteristic (AUROC), profiling the detectors on the whole spectrum of the thresholds. AUROC ranges from 0.0 to 1.0, mathematically denoting the probability of a random machine-generated text having a higher predicted probability of being machine-generated than a random human-written text. Typically, an AUROC of 0.5 indicates a random classifier and an AUROC of 1.0 indicates a perfect classifier. We also report the relative improvement calculated by $(new - old)/(1.0 - old)$, which represents how much improvement has been made relative to the maximum possible improvement.

## B  Analytical Solution of Conditional Probability Curvature

The sample mean $\tilde{\mu}$ in Eq. 4 represents the cross-entropy of distribution $q_\varphi(\tilde{x}|x)$ and $p_\theta(\tilde{x}|x)$. By leveraging the conditional independence of each token, we can calculate the expectation analytically as

$$\tilde{\mu} = \mathbb{E}_{\tilde{x} \sim q_\varphi(\tilde{x}|x)} \left[ \log p_\theta(\tilde{x}|x) \right] = \sum_{\tilde{x}} q_\varphi(\tilde{x}|x) \log p_\theta(\tilde{x}|x)$$
$$= \sum_j \sum_{\tilde{x}_j} q_\varphi(\tilde{x}_j|x_{<j}) \log p_\theta(\tilde{x}_j|x_{<j}) = \sum_j \tilde{\mu}_j, \tag{6}$$

where $\tilde{\mu}_j$ denotes the sample mean on the $j$-th token. The summation over $\tilde{x}_j$ is computed by enumerating the tokens in the vocabulary, which can be exactly calculated.

The sample variance $\tilde{\sigma}^2$ in Eq. 4 can also be calculated analytically

$$\tilde{\sigma}^2 = \mathbb{E}_{\tilde{x} \sim q_\varphi(\tilde{x}|x)} \left[ (\log p_\theta(\tilde{x}|x) - \tilde{\mu})^2 \right] = \sum_{\tilde{x}} q_\varphi(\tilde{x}|x)(\log p_\theta(\tilde{x}|x) - \tilde{\mu})^2$$
$$= \sum_j \left( \sum_{\tilde{x}_j} q_\varphi(\tilde{x}_j|x_{<j}) \log^2 p_\theta(\tilde{x}_j|x_{<j}) - \tilde{\mu}_j^2 \right), \tag{7}$$

where the summation over $\tilde{x}_j$ can also be calculated exactly by enumerating the tokens in the vocabulary. Empirically, the analytical solution achieves a detection accuracy almost identical to the sampling approximation with 10,000 samples (our default setting) but further accelerates the detection process by about 10%.

## C  EXPERIMENTAL SETTINGS

### C.1  OPEN-SOURCE AND PROPRIETARY MODELS

| Model | Model File/Service | Parameters | Training Corpus |
|---|---|---|---|
| mGPT (Shliazhko et al., 2022) | sberbank-ai/mGPT | 1.3B | Wikipedia and Colossal Clean Crawled corpus. |
| GPT-2 (Radford et al., 2019) | gpt2-xl | 1.5B | English WebText without Wikipedia. |
| PubMedGPT (Bolton et al., 2022) | stanford-crfm/pubmedgpt | 2.7B | Biomedical abstracts and papers from the Pile. |
| OPT-2.7 (Zhang et al., 2022) | facebook/opt-2.7b | 2.7B | A larger dataset including the Pile. |
| Neo-2.7 (Black et al., 2021) | EleutherAI/gpt-neo-2.7B | 2.7B | The Pile (Gao et al., 2020). |
| GPT-J (Wang & Komatsuzaki, 2021) | EleutherAI/gpt-j-6B | 6B | The Pile (Gao et al., 2020). |
| BLOOM-7.1 (Scao et al., 2022) | bigscience/bloom-7b1 | 7.1B | ROOTS corpus (Laurençon et al., 2022). |
| OPT-13 (Zhang et al., 2022) | facebook/opt-13b | 13B | A larger dataset including the Pile. |
| Llama-13 (Touvron et al., 2023a) | huggyllama/llama-13b | 13B | CommonCrawl, C4, Github, Wikipedia, Books, ... |
| Llama2-13 (Touvron et al., 2023b) | TheBloke/Llama-2-13B-fp16 | 13B | A new mix of publicly available online data. |
| NeoX (Black et al., 2022) | EleutherAI/gpt-neox-20b | 20B | The Pile (Gao et al., 2020). |
| GPT-3 (Brown et al., 2020) | OpenAI/davinci | 175B | CommonCrawl, WebText, English Wikipedia, ... |
| ChatGPT (OpenAI, 2022) | OpenAI/gpt-3.5-turbo | 175B | CommonCrawl, WebText, English Wikipedia, ... |
| GPT-4 (OpenAI, 2023) | OpenAI/gpt-4 | NA | NA |

Table 4: Details of the source models that is used to produce machine-generated text.

We assess the performance of our methodologies using text generations sourced from various models, as outlined in Table 4. These models are arranged in order of their parameter count, with those having fewer than 20 billion parameters being run locally on a Tesla A100 GPU (80G). For models with over 6 billion parameters, we employ half-precision (float16), otherwise, we use full-precision (float32).

In the case of larger models like GPT-3, ChatGPT, and GPT-4, we utilize the OpenAI API for the evaluations. Additionally, we provide information about the training corpus associated with each model, which we believe is pertinent for understanding the detection accuracy of different sampling and scoring models when applied to text generations originating from diverse source models, domains, and languages.

### C.2  SETTINGS FOR GPT-3, CHATGPT, AND GPT-4

To compile our test set, we generate 150 samples for each dataset (among XSum, WritingPrompts, and PubMedQA) and each source model by calling the OpenAI service [3]. Specifically, for GPT-3, we requested text completions for a 30-token prefix, while for ChatGPT and GPT-4, we request chat completions with predefined instructions as follows.

ChatGPT and GPT-4 function in a conversational manner, serving as assistants to execute user instructions. In the context of our experiment, we task these models with adopting the roles of professional News, Fiction, and Technical writers for the generation of News articles, stories, and answers, respectively. To encourage the production of content that is both unpredictable and creatively diverse, we utilize a temperature setting of $0.8$.

Specifically, we initiate the generation process by sending the following messages to the service.

Message for **XSum**:

```
[
    {'role': 'system', 'content': 'You are a News writer.'},
    {'role': 'user', 'content': 'Please write an article with about 150
        ↪ words starting exactly with: <prefix>'},
]
```

The <prefix> could be like "Maj Richard Scott, 40, is accused of driving at speeds of up to 95mph (153km/h) in bad weather", and the response is supposed to start with it.

Message for **WritingPrompts**:

---

[3] https://openai.com/blog/openai-api

```
1  [
2      {'role': 'system', 'content': 'You are a Fiction writer.'},
3      {'role': 'user', 'content': 'Please write an article with about 150
          ↪ words starting exactly with: <prefix>'},
4  ]
```

The <prefix> could be like "A man invents time travel in order to find a cure for his sick wife and succeeds, only to find out", and the response is supposed to start with it.

Message for **PubMedQA**:

```
1  [
2      {'role': 'system', 'content': 'You are a Technical writer.'},
3      {'role': 'user', 'content': 'Please answer the question in about 50
          ↪ words. <prefix>'},
4  ]
```

The <prefix> could be like "Question: Is an advance care planning model feasible in community palliative care? Answer:" and the response is supposed to answer the question directly.

| Dataset | Method | Source Model | | | | | |
| --- | --- | --- | --- | --- | --- | --- | --- |
| | | GPT-2 | OPT-2.7 | Neo-2.7 | GPT-J | NeoX | Avg. |
| XSum | Likelihood | 0.8638 | 0.8600 | 0.8609 | 0.8101 | 0.7604 | 0.8310 |
| | Entropy | 0.5835 | 0.5071 | 0.5712 | 0.5705 | 0.6035 | 0.5671 |
| | LogRank | 0.8918 | 0.8839 | 0.8949 | 0.8407 | 0.7939 | 0.8610 |
| | LRR | 0.9179 | 0.8867 | 0.9190 | 0.8592 | 0.8205 | 0.8807 |
| | DNA-GPT ◇ | 0.8548 | 0.8168 | 0.8197 | 0.7586 | 0.7167 | 0.7933 |
| | NPR ◇ | 0.9891* | 0.9681* | 0.9929* | 0.9566 | 0.9311 | 0.9676 |
| | DetectGPT ◇ | 0.9875 | 0.9621 | 0.9914 | 0.9632* | 0.9398* | 0.9688* |
| | Fast-DetectGPT | **0.9930** | **0.9803** | 0.9885 | **0.9771** | **0.9703** | **0.9818** |
| | *(Diff)* | *0.0055* | *0.0182* | *-0.0029* | *0.0139* | *0.0305* | *0.0130* |
| | DetectGPT (fixed) ◇ | 0.9180 | 0.8868 | 0.9914 | 0.8830 | 0.8682 | 0.9095 |
| | Fast-DetectGPT (fixed) | 0.9742 | 0.9444 | **0.9965** | 0.9335 | 0.9033 | 0.9504 |
| | *(Diff)* | *0.0563* | *0.0576* | *0.0051* | *0.0505* | *0.0351* | *0.0409* |
| SQuAD | Likelihood | 0.9077 | 0.8839 | 0.8585 | 0.7943 | 0.6977 | 0.8284 |
| | Entropy | 0.5791 | 0.5119 | 0.5581 | 0.5643 | 0.6056 | 0.5638 |
| | LogRank | 0.9454 | 0.9203 | 0.9054 | 0.8471 | 0.7545 | 0.8745 |
| | LRR | 0.9773 | 0.9597 | 0.9610 | 0.9244 | 0.8600 | 0.9365 |
| | DNA-GPT ◇ | 0.9094 | 0.8934 | 0.8589 | 0.8069 | 0.7525 | 0.8442 |
| | NPR ◇ | 0.9965* | 0.9853* | 0.9789 | 0.9108 | 0.8175 | 0.9378 |
| | DetectGPT ◇ | 0.9914 | 0.9763 | 0.9625 | 0.8738 | 0.7916 | 0.9191 |
| | Fast-DetectGPT | **0.9990** | **0.9949** | 0.9956* | **0.9853** | **0.9617** | **0.9873** |
| | *(Diff)* | *0.0076* | *0.0186* | *0.0331* | *0.1116* | *0.1702* | *0.0682* |
| | DetectGPT (fixed) ◇ | 0.7382 | 0.7530 | 0.9625 | 0.7882 | 0.7709 | 0.8026 |
| | Fast-DetectGPT (fixed) | 0.9824 | 0.9762 | **0.9990** | 0.9584* | 0.9379* | 0.9708* |
| | *(Diff)* | *0.2442* | *0.2232* | *0.0365* | *0.1703* | *0.1669* | *0.1682* |
| Writing Prompts | Likelihood | 0.9661 | 0.9451 | 0.9505 | 0.9396 | 0.9256 | 0.9454 |
| | Entropy | 0.3895 | 0.4299 | 0.3400 | 0.3668 | 0.3908 | 0.3834 |
| | LogRank | 0.9782 | 0.9628 | 0.9675 | 0.9577 | 0.9454 | 0.9623 |
| | LRR | 0.9850 | 0.9740 | 0.9766 | 0.9702 | 0.9573 | 0.9726 |
| | DNA-GPT ◇ | 0.9431 | 0.9288 | 0.9283 | 0.9026 | 0.8786 | 0.9163 |
| | NPR ◇ | **0.9987** | 0.9962* | 0.9930 | 0.9825 | 0.9708 | 0.9882 |
| | DetectGPT ◇ | 0.9962 | 0.9891 | 0.9852 | 0.9688 | 0.9516 | 0.9782 |
| | Fast-DetectGPT | 0.9982* | **0.9972** | 0.9980* | **0.9974** | **0.9941** | **0.9970** |
| | *(Diff)* | *0.0020* | *0.0081* | *0.0129* | *0.0285* | *0.0424* | *0.0188* |
| | DetectGPT (fixed) ◇ | 0.8989 | 0.8772 | 0.9852 | 0.9014 | 0.8809 | 0.9087 |
| | Fast-DetectGPT (fixed) | 0.9937 | 0.9509 | **0.9996** | 0.9858* | 0.9801* | 0.9820* |
| | *(Diff)* | *0.0948* | *0.0738* | *0.0145* | *0.0844* | *0.0992* | *0.0733* |

Table 5: Details of the main results in Table 2: Zero-shot detection on five source models, where Fast-DetectGPT consistently outperforms DetectGPT in terms of detection accuracy (in AUROC). We run DetectGPT and NPR with default 100 perturbations and run DNA-GPT with a truncate-ratio of 0.5 and 10 prefix completions per passage. Methods marked with "*(fixed)*" use the fixed models to detect texts from different sources (the black-box setting), where DetectGPT uses T5-3B/Neo-2.7 as the perturbation/scoring models and Fast-DetectGPT uses GPT-J/Neo-2.7 as the sampling/scoring models. The "*(Diff)*" rows indicate the AUROC improvement upon DetectGPT baselines. "*" denotes the second-best AUROC. ◇ – Methods call models a hundred times, thus consuming much higher computational resources.

# D    ADDITIONAL RESULTS

## D.1    ZERO-SHOT DETECTION ON ADDITIONAL OPEN-SOURCE MODELS

| Dataset | Method | Source Model | | | | |
|---|---|---|---|---|---|---|
| | | BLOOM-7.1 | OPT-13 | Llama-13 | Llama2-13 | Avg. |
| XSum | Likelihood | 0.8500 | 0.8105 | 0.6121 | 0.6550 | 0.7319 |
| | Entropy | 0.6642 | 0.5251 | 0.6731 | 0.6029 | 0.6163 |
| | LogRank | 0.9018 | 0.8369 | 0.6542 | 0.6911 | 0.7710 |
| | LRR | 0.9412 | 0.8344 | 0.7327 | 0.7351 | 0.8109 |
| | NPR | **0.9931** | 0.9283* | 0.8031 | 0.8212* | 0.8864* |
| | DetectGPT | 0.9912* | 0.9268 | 0.8147* | 0.8106 | 0.8858 |
| | Fast-DetectGPT | 0.9890 | **0.9721** | **0.9473** | **0.9346** | **0.9607** |
| | *(Diff)* | *-0.0021* | *0.0452* | *0.1325* | *0.1240* | *0.0749* |
| | DetectGPT (fixed) | 0.7365 | 0.8713 | 0.6869 | 0.6848 | 0.7449 |
| | Fast-DetectGPT (fixed) | 0.8886 | 0.9207 | 0.7306 | 0.6968 | 0.8092 |
| | *(Diff)* | *0.1521* | *0.0495* | *0.0436* | *0.0121* | *0.0643* |
| SQuAD | Likelihood | 0.8619 | 0.8220 | 0.5113 | 0.5000 | 0.6738 |
| | Entropy | 0.6199 | 0.5517 | 0.6534 | 0.6636 | 0.6221 |
| | LogRank | 0.9157 | 0.8645 | 0.5589 | 0.5457 | 0.7212 |
| | LRR | 0.9678 | 0.9228 | 0.7262 | 0.7036 | 0.8301 |
| | NPR | 0.9730* | 0.9351 | 0.5332 | 0.5448 | 0.7465 |
| | DetectGPT | 0.9510 | 0.9110 | 0.5204 | 0.5507 | 0.7333 |
| | Fast-DetectGPT | **0.9953** | **0.9893** | **0.8717** | **0.8772** | **0.9333** |
| | *(Diff)* | *0.0443* | *0.0782* | *0.3513* | *0.3265* | *0.2001* |
| | DetectGPT (fixed) | 0.6359 | 0.7596 | 0.5588 | 0.5488 | 0.6258 |
| | Fast-DetectGPT (fixed) | 0.9588 | 0.9590* | 0.8028* | 0.7627* | 0.8708* |
| | *(Diff)* | *0.3228* | *0.1994* | *0.2440* | *0.2138* | *0.2450* |
| WritingPrompts | Likelihood | 0.9368 | 0.9400 | 0.8692 | 0.8737 | 0.9049 |
| | Entropy | 0.4876 | 0.4096 | 0.4831 | 0.4702 | 0.4626 |
| | LogRank | 0.9612 | 0.9578 | 0.9047 | 0.9069 | 0.9327 |
| | LRR | 0.9811 | 0.9650 | 0.9326* | 0.9306* | 0.9523* |
| | NPR | 0.9909* | 0.9850* | 0.9184 | 0.9003 | 0.9487 |
| | DetectGPT | 0.9829 | 0.9701 | 0.8596 | 0.8396 | 0.9130 |
| | Fast-DetectGPT | **0.9983** | **0.9953** | **0.9892** | **0.9939** | **0.9942** |
| | *(Diff)* | *0.0154* | *0.0253* | *0.1296* | *0.1543* | *0.0811* |
| | DetectGPT (fixed) | 0.7933 | 0.8695 | 0.7455 | 0.7532 | 0.7904 |
| | Fast-DetectGPT (fixed) | 0.9779 | 0.9367 | 0.8925 | 0.9085 | 0.9289 |
| | *(Diff)* | *0.1846* | *0.0672* | *0.1471* | *0.1553* | *0.1385* |

Table 6: Addition to the main results in Table 5: Zero-shot detection on more source models.

We extend our evaluation to include several popular open-source LLMs, including BLOOM 7.1B, OPT 13B, Llama 13B, and Llama2 13B. In the white-box setting, Fast-DetectGPT exhibits an average relative improvement of 76.1% when compared to DetectGPT. This outcome aligns with the 74.7% average relative improvement observed across the five models presented in Table 5, underscoring the consistent performance of Fast-DetectGPT across diverse source models.

However, in the black-box setting, Fast-DetectGPT demonstrates an average relative improvement of 53.4% compared to DetectGPT. This figure is lower than the 74.5% average relative improvement seen across the five models in the main table. We suspect that the reduced improvement observed in these source models relate to the potential mismatch between the scoring model Neo-2.7 and the source models. It is conceivable that the training corpus used by the former may have limited overlap with the training corpus utilized by the latter according to Table 4. These findings underscore the challenges associated with identifying universally effective scoring models in the black-box setting.

## D.2 ZERO-SHOT DETECTION ON GPT-3 GENERATIONS

| Method | XSum | WritingPrompts | PubMedQA | Avg. |
|---|---|---|---|---|
| RoBERTa-base | 0.8986 | 0.9282 | 0.6370 | 0.8212 |
| RoBERTa-large | 0.9325 | 0.9113 | 0.6894 | 0.8444 |
| GPTZero | 0.4860 | 0.6009 | 0.4246 | 0.5038 |
| Likelihood (GPT-3) ◇ | 0.76 | 0.87 | 0.64 | 0.76 |
| DetectGPT (T5-11B/GPT-3) ◇ | 0.84 | 0.87 | **0.84** | 0.85 |
| Likelihood (Neo-2.7) | 0.8307 | 0.8496 | 0.5668 | 0.7490 |
| Entropy (Neo-2.7) | 0.3923 | 0.3049 | 0.5358 | 0.4110 |
| LogRank(Neo-2.7) | 0.8096 | 0.8320 | 0.5527 | 0.7314 |
| LRR (Neo-2.7) | 0.6687 | 0.7410 | 0.4917 | 0.6338 |
| DNA-GPT (Neo-2.7) | 0.8209 | 0.8354 | 0.5761 | 0.7441 |
| NPR (T5-11B/Neo-2.7) | 0.8032 | 0.7847 | 0.6342 | 0.7407 |
| DetectGPT (T5-11B/GPT-2) | 0.8043 | 0.7699 | 0.6915 | 0.7552 |
| DetectGPT (T5-11B/Neo-2.7) | 0.8455 | 0.7818 | 0.6977 | 0.7750 |
| DetectGPT (T5-11B/GPT-J) | 0.8261 | 0.7666 | 0.6644 | 0.7524 |
| Fast-DetectGPT (GPT-J/GPT-2) | 0.9137 | 0.9533* | 0.7589* | **0.8753** |
| Fast-DetectGPT (GPT-J/Neo-2.7) | **0.9396** | 0.9492 | 0.7225 | 0.8704* |
| Fast-DetectGPT (GPT-J/GPT-J) | 0.9329* | **0.9568** | 0.6664 | 0.8520 |

Table 7: Detection of *GPT-3* generations, evaluated in AUROC. Fast-DetectGPT in the black-box settings (using local models) significantly outperforms DetectGPT in both the black-box setting and the white-box setting (using GPT-3) on News (XSum) and story (WritingPrompts). Fast-DetectGPT uses 6B GPT-J to generate samples and models from 1.5B GPT-2 to 6B GPT-J to score samples, while DetectGPT uses 11B T5 to generate perturbations and models from 1.5B GPT-2 to 6B GPT-J, and GPT-3 service to score them. ◇ – we report the official scores from Mitchell et al. (2023) instead of rerunning the experiments after confirming the consistency on RoBERTa-base/large.

Table 7 presents a comparison between Fast-DetectGPT, zero-shot DetectGPT, and supervised RoBERTa-based classifiers for the detection of GPT-3 generations. In contrast to DetectGPT, which employs the OpenAI API to assess perturbations, we utilize delegate models (specifically, GPT-2, Neo-2.7, and GPT-J) to identify GPT-3 generations.

Fast-DetectGPT outperforms supervised RoBERTa-base, RoBERTa-large, and GPTZero classifiers, achieving higher detection accuracy across the three datasets. On average, it improves the AUROC by 0.0310 AUROC (a relative increase of 20%). Conversely, DetectGPT in the white-box setting (using T5-11B/GPT-3) achieves superior detection accuracy on PubMedQA but lags behind on XSum and WritingPrompt compared to RoBERTa-large. In the black-box setting (T5-11B/Neo-2.7), DetectGPT experiences a significant reduction in detection accuracy, averaging 0.0750 AUROC less. These findings emphasize that *Fast-DetectGPT, operating in the black-box setting, offers a competitive alternative to supervised detectors and DetectGPT in the white-box setting*.

When comparing the results on GPT-3 and ChatGPT, it becomes apparent that Fast-DetectGPT performs significantly better on ChatGPT. We speculate that this discrepancy may relate to factors such as instruction-tuning (Wei et al., 2021) and human-feedback reinforcement learning (HFRL) (Ouyang et al., 2022), which are employed in fine-tuning large language models and may skew the model toward high-probability tokens.

# E  ABLATION STUDY

| Method | XSum | | | SQuAD | | | WritingPrompts | | | Avg. |
|---|---|---|---|---|---|---|---|---|---|---|
| | GPT-2 | Neo-2.7 | GPT-J | GPT-2 | Neo-2.7 | GPT-J | GPT-2 | Neo-2.7 | GPT-J | |
| Fast-DetectGPT (*/*) | 0.9930 | 0.9885 | **0.9771** | 0.9990 | 0.9956 | **0.9853** | 0.9982 | 0.9980 | **0.9974** | 0.9925 |
| Fast-DetectGPT (GPT-2/*) | 0.9930 | 0.9918 | 0.9534 | 0.9990 | 0.9728 | 0.8785 | 0.9982 | 0.9954 | 0.9868 | 0.9743 |
| Fast-DetectGPT (Neo-2.7/*) | 0.9778 | 0.9885 | 0.9153 | 0.9977 | 0.9956 | 0.9212 | 0.9994 | 0.9980 | 0.9861 | 0.9755 |
| Fast-DetectGPT (GPT-J/*) | **0.9960** | **0.9965** | **0.9771** | **1.0000** | **0.9990** | **0.9853** | **0.9999** | **0.9996** | **0.9974** | **0.9945** |

Table 8: Impact of reference model, where "*/*" indicates that we use the source model to generate reference samples and score the log probability, while "GPT-J/*" indicates that we use GPT-J to generate the samples and the source model to score.

**Sampling Model Ablation.** We investigate the influence of the choice of the sampling model, as summarized in Table 8. Remarkably, when GPT-J is employed as the sampling model, Fast-DetectGPT attains the highest average detection accuracy. In comparison to Fast-DetectGPT utilizing the source model for sampling, the utilization of GPT-J enhances accuracy by an average of 0.0020 AUROC, representing a relative improvement of 27% across all three datasets and the three models under consideration. These findings indicate that employing a more robust sampling model has the potential to further augment the performance of Fast-DetectGPT in the white-box setting.

**Normalization Ablation.** Normalization based on the standard deviation has previously been proposed as an additional technique within DetectGPT. In our study, we integrate this normalization as a default component of the conditional probability curvature metric for two principal reasons. Firstly, normalization exerts a significant influence on detection accuracy, resulting in an average AUROC improvement of 0.0172, equivalent to a relative increase of 36% for DetectGPT. In the case of Fast-DetectGPT, normalization enhances the average AUROC by 0.0020, corresponding to a 10% relative improvement. Secondly, after normalization, the distributions of the curvatures for different models across various datasets become more directly comparable. Without normalization, these distributions exhibit variations in width, ranging from narrow to wide, depending on the variance of the generated samples.

**Entropy Ablation.** Among the total 75% relative improvement, 10% is attributed to normalization by $\tilde{\sigma}$, while the remaining 65% stems from the contribution of the numerator $\log p_\theta(x|x) - \tilde{\mu}$. The entropy $-\tilde{\mu}$ plays a crucial role in achieving high detection accuracy in Fast-DetectGPT.

An intuitive elucidation of the significance of entropy lies in the substantial variance observed in the $\log p_\theta(x_j|x_{<j})$ values for different tokens $x_j$ across diverse contexts $x_{<j}$. This variability introduces instability in the statistical measures employed for detection. Conversely, the expectation $\tilde{\mu}_j$ establishes a dynamic probability baseline for each token. Consequently, the subtraction of $\log p\theta(x_j|x_{<j})$ and $\tilde{\mu}_j$ serves to mitigate the variance of the log-likelihood, yielding a more stable statistic that proves resilient to token or context fluctuations.

In a specific experiment involving ChatGPT generations for XSum, the average standard deviation of token-level log-likelihood is 2.1893, while the average standard deviation of token-level entropy is 1.6090. Conversely, the average standard deviation resulting from their addition is 1.6342, representing a significant reduction from the initial 2.1893.

# F    Additional Analysis

## F.1    Robustness against Decoding Strategies

| Method | Top-$p$ ($p=0.96$) | | | | Top-$k$ ($k=40$) | | | | Temperature ($T=0.8$) | | | |
|---|---|---|---|---|---|---|---|---|---|---|---|---|
| | XSum | SQuAD | WritingP | Avg. | XSum | SQuAD | WritingP | Avg. | XSum | SQuAD | WritingP | Avg. |
| Likelihood | 0.9126 | 0.9045 | 0.9781 | 0.9317 | 0.8624 | 0.8612 | 0.9608 | 0.8948 | 0.973 | 0.9647 | 0.9962 | 0.9780 |
| Entropy | 0.5287 | 0.5273 | 0.3255 | 0.4605 | 0.5538 | 0.5523 | 0.3632 | 0.4898 | 0.4854 | 0.4942 | 0.2522 | 0.4106 |
| LogRank | 0.9293 | 0.9324 | 0.9853 | 0.9490 | 0.8946 | 0.9047 | 0.9757 | 0.9250 | 0.9844 | 0.9821 | 0.9978 | 0.9881 |
| LRR | 0.9223 | 0.9600 | 0.9836 | 0.9553 | 0.9173 | 0.9566 | 0.9827 | 0.9522 | 0.9826 | 0.9923 | 0.9903 | 0.9884 |
| NPR | 0.9789 | 0.9511 | 0.9901 | 0.9734 | 0.9726 | 0.945 | 0.9912 | 0.9696 | 0.9892 | 0.9710 | 0.9897 | 0.9833 |
| DetectGPT | 0.9778 | 0.9359 | 0.9807 | 0.9648 | 0.9708 | 0.9247 | 0.9797 | 0.9584 | 0.9830 | 0.9362 | 0.9745 | 0.9646 |
| Fast-Detect | **0.9975** | **0.9976** | **0.9994** | **0.9982** | **0.9871** | **0.9914** | **0.9977** | **0.9921** | **0.9998** | **0.9996** | **0.9992** | **0.9995** |
| *(Diff)* | *0.0197* | *0.0617* | *0.0188* | *0.0334* | *0.0164* | *0.0667* | *0.0180* | *0.0337* | *0.0168* | *0.0634* | *0.0247* | *0.0350* |
| DetectGPT (fixed) | 0.9476 | 0.8506 | 0.9377 | 0.9120 | 0.9158 | 0.8202 | 0.9181 | 0.8847 | 0.9717 | 0.9026 | 0.9522 | 0.9422 |
| Fast-Detect (fixed) | 0.9889 | 0.9942 | 0.9945 | 0.9925 | 0.9642 | 0.9790 | 0.9864 | 0.9765 | 0.9988 | 0.9989 | 0.9984 | 0.9987 |
| *(Diff)* | *0.0413* | *0.1436* | *0.0568* | *0.0806* | *0.0484* | *0.1587* | *0.0683* | *0.0918* | *0.0271* | *0.0963* | *0.0462* | *0.0565* |
| **Method** | Top-$p$ ($p=0.90$) | | | | Top-$k$ ($k=30$) | | | | Temperature ($T=0.6$) | | | |
| Likelihood | 0.9592 | 0.9495 | 0.9924 | 0.9670 | 0.9010 | 0.8922 | 0.9754 | 0.9229 | 0.9964 | 0.9954 | 0.9999 | 0.9972 |
| Entropy | 0.4985 | 0.4990 | 0.2881 | 0.4285 | 0.532 | 0.5374 | 0.3359 | 0.4684 | 0.4171 | 0.3718 | 0.1146 | 0.3012 |
| LogRank | 0.9701 | 0.9684 | 0.9951 | 0.9779 | 0.9309 | 0.9338 | 0.9863 | 0.9503 | 0.9990 | 0.9987 | **1.0000** | 0.9992 |
| Fast-Detect | **0.9997** | **0.9998** | **0.9996** | **0.9997** | **0.9941** | **0.9955** | **0.9988** | **0.9961** | **0.9998** | **0.9999** | 0.9991 | **0.9996** |

Table 9: Impact of *decoding strategies*, where the machine-generated texts are produced by sampling with top-$p$, top-$k$, and temperature. The report AUROC is the average over the five models in Table 10.

| Method | Top-$p$ ($p=0.96$) | | | | | Top-$k$ ($k=40$) | | | | | Temperature ($T=0.8$) | | | | |
|---|---|---|---|---|---|---|---|---|---|---|---|---|---|---|---|
| | GPT2 | OPT2.7 | Neo2.7 | GPTJ | NeoX | GPT2 | OPT2.7 | Neo2.7 | GPTJ | NeoX | GPT2 | OPT2.7 | Neo2.7 | GPTJ | NeoX |
| **XSum** | | | | | | | | | | | | | | | |
| Likelihood | .9234 | .9308 | .931 | .9042 | .8733 | .8813 | .8918 | .8873 | .8401 | .8114 | .9781 | .982 | .9862 | .9679 | .9511 |
| Entropy | .5541 | .4665 | .5303 | .531 | .5614 | .5743 | .4843 | .5601 | .5588 | .5917 | .5022 | .4289 | .4669 | .4837 | .5453 |
| LogRank | .9414 | .9422 | .9502 | .922 | .8906 | .9122 | .9173 | .9227 | .8756 | .8454 | .9883 | .9885 | .9939 | .9817 | .9694 |
| LRR | .9509 | .9252 | .9478 | .9203 | .8672 | .9432 | .9254 | .9452 | .9045 | .8683 | .9889 | .9823 | .991 | .9806 | .9702 |
| NPR | .9909 | .9841 | .9982 | .973 | .9486 | .987 | .9801 | .9938 | .9606 | .9416 | .9955 | .9916 | .9976 | .9853 | .976 |
| DetectGPT | .9875 | .9793 | .9961 | .9762 | .95 | .9869 | .9707 | .9919 | .9619 | .9424 | .9928 | .9856 | .9953 | .979 | .9622 |
| Fast-DetectGPT | .9994 | .9965 | .9988 | .997 | .9958 | .9954 | .9867 | .9938 | .9826 | .9773 | .9999 | .9999 | .9997 | .9999 | .9997 |
| *(Diff)* | *.0119* | *.0172* | *.0028* | *.0208* | *.0458* | *.0085* | *.0160* | *.0018* | *.0207* | *.0349* | *.0071* | *.0143* | *.0044* | *.0209* | *.0376* |
| DetectGPT(fixed) | .9399 | .9438 | .9961 | .9367 | .9214 | .9143 | .9026 | .9919 | .8846 | .8854 | .9722 | .9635 | .9953 | .9627 | .9646 |
| Fast-Detect(fixed) | .9953 | .9861 | .9997 | .9856 | .9779 | .9805 | .9613 | .9979 | .9499 | .9311 | .9978 | .999 | .9999 | .9991 | .998 |
| *(Diff)* | *.0554* | *.0423* | *.0036* | *.0489* | *.0565* | *.0662* | *.0587* | *.0060* | *.0654* | *.0457* | *.0256* | *.0355* | *.0046* | *.0364* | *.0333* |
| **SQuAD** | | | | | | | | | | | | | | | |
| Likelihood | .961 | .944 | .9214 | .8838 | .8122 | .9393 | .9072 | .8926 | .8351 | .7317 | .9906 | .987 | .9792 | .9572 | .9094 |
| Entropy | .5369 | .4736 | .539 | .5277 | .5593 | .552 | .5203 | .5457 | .5441 | .5992 | .5132 | .4508 | .4924 | .4882 | .5263 |
| LogRank | .9792 | .9657 | .9535 | .9156 | .8482 | .9692 | .9423 | .9385 | .8842 | .7895 | .9972 | .9959 | .994 | .9798 | .9434 |
| LRR | .9865 | .981 | .981 | .9507 | .9009 | .9898 | .9782 | .9815 | .945 | .8886 | .9991 | .9973 | .9992 | .9911 | .9748 |
| NPR | .9955 | .9934 | .9897 | .9335 | .8436 | .9962 | .99 | .9877 | .9251 | .826 | .9988 | .9963 | .9931 | .9684 | .8981 |
| DetectGPT | .994 | .9838 | .9781 | .9054 | .8182 | .9928 | .9804 | .9709 | .8819 | .7977 | .9959 | .9819 | .9778 | .9123 | .8132 |
| Fast-DetectGPT | 1 | .9997 | .9998 | .9981 | .9903 | .9986 | .9973 | .998 | .9928 | .9706 | 1 | 1 | 1 | .9994 | .9986 |
| *(Diff)* | *.0060* | *.0159* | *.0218* | *.0927* | *.1722* | *.0057* | *.0169* | *.0271* | *.1109* | *.1729* | *.0041* | *.0180* | *.0222* | *.0871* | *.1854* |
| DetectGPT(fixed) | .8066 | .7857 | .9781 | .8462 | .8365 | .7815 | .763 | .9709 | .8128 | .7729 | .8936 | .862 | .9778 | .8897 | .8898 |
| Fast-Detect(fixed) | .9984 | .995 | 1 | .9933 | .9845 | .9893 | .984 | .9997 | .9709 | .9509 | 1 | .9999 | 1 | .9984 | .9961 |
| *(Diff)* | *.1918* | *.2093* | *.0219* | *.1471* | *.1480* | *.2078* | *.2210* | *.0288* | *.1581* | *.1780* | *.1064* | *.1379* | *.0222* | *.1087* | *.1063* |
| **WritingPrompts** | | | | | | | | | | | | | | | |
| Likelihood | .9889 | .9736 | .9806 | .9764 | .9711 | .9777 | .9555 | .9677 | .9566 | .9463 | .9983 | .9949 | .9971 | .996 | .9945 |
| Entropy | .3236 | .3789 | .2798 | .3212 | .3238 | .3601 | .4149 | .3183 | .3529 | .3696 | .2511 | .3033 | .219 | .2252 | .2625 |
| LogRank | .9931 | .9827 | .987 | .9853 | .9785 | .9867 | .9719 | .9807 | .9735 | .9656 | .9991 | .9972 | .9984 | .9977 | .9969 |
| LRR | .9906 | .9854 | .9857 | .9839 | .9724 | .9901 | .9835 | .9858 | .9834 | .9709 | .9961 | .9917 | .9914 | .9887 | .9837 |
| NPR | .9984 | .9964 | .9944 | .9855 | .9758 | .9988 | .9955 | .9971 | .9888 | .9759 | .9985 | .9953 | .993 | .9836 | .9781 |
| DetectGPT | .9969 | .9898 | .9926 | .9714 | .9526 | .9965 | .9868 | .992 | .971 | .9521 | .9932 | .985 | .9842 | .9656 | .9447 |
| Fast-DetectGPT | .9997 | .9999 | .9998 | .9998 | .9981 | .9978 | .9973 | .999 | .9987 | .9957 | .9997 | .9999 | .9992 | .9991 | .9984 |
| *(Diff)* | *.0028* | *.0102* | *.0071* | *.0284* | *.0455* | *.0012* | *.0105* | *.0070* | *.0277* | *.0436* | *.0065* | *.0149* | *.0150* | *.0335* | *.0537* |
| DetectGPT(fixed) | .9297 | .9177 | .9926 | .9299 | .9187 | .9115 | .8906 | .992 | .9052 | .8914 | .9385 | .9451 | .9842 | .9535 | .9398 |
| Fast-Detect(fixed) | .9982 | .9825 | .9999 | .9974 | .9945 | .9947 | .9616 | .9998 | .9901 | .986 | .9995 | .9959 | .9997 | .9983 | .9987 |
| *(Diff)* | *.0685* | *.0648* | *.0073* | *.0676* | *.0758* | *.0832* | *.0710* | *.0079* | *.0849* | *.0946* | *.0610* | *.0508* | *.0155* | *.0448* | *.0588* |

Table 10: Detailed results on decoding strategies.

Machine systems may use different decoding strategies such as top-$k$ sampling, top-$p$ (Nucleus) sampling, and sampling with a temperature $T$. We experiment on the five models and three datasets

used in Table 2, sampling with the three strategies with $k = 40$, $p = 0.96$, and $T = 0.8$. Fast-DetectGPT in the white-box setting obtains the best accuracy on the three sampling strategies, outperforming DetectGPT by relative 95% on Top-$p$ sampling, relative 81% on Top-$k$ sampling, and relative 99% on sampling with a temperature, as shown in Table 9. In the black-box setting, Fast-DetectGPT outperforms DetectGPT by relatively 92%, 80%, and 98% on the three decoding strategies, respectively. These results demonstrate that Fast-DetectGPT works consistently in detecting texts produced by different decoding strategies.

To elucidate the trajectory of detection accuracy concerning variations in sampling hyperparameters, we conducted additional experiments with values set to $p = 0.90$, $k = 30$, and $T = 0.6$. As indicated in the lower segment of Table 9, reducing the values of $p$, $k$, and $T$ enhances the determinism of generated samples, facilitating easier detection and consequently yielding significantly higher AUROCs.

## F.2 ROBUSTNESS UNDER PARAPHRASING ATTACK

| Method | Paraphrasing Attack | | | | Decoherence Attack | | | |
|---|---|---|---|---|---|---|---|---|
| | XSum | Writing | PubMed | Avg. | XSum | Writing | PubMed | Avg. |
| RoBERTa-base | 0.8103 | 0.5368 | 0.5962 | 0.6477 | 0.5778 | 0.5629 | 0.5392 | 0.5600 |
| RoBERTa-large | 0.7532 | 0.4619 | 0.6187 | 0.6113 | 0.4945 | 0.3849 | 0.5520 | 0.4771 |
| Likelihood(Neo-2.7) | 0.8521 | 0.8691 | 0.7029 | 0.8080 | 0.7393 | 0.9240 | 0.7757 | 0.8130 |
| Entropy(Neo-2.7) | 0.4508 | 0.3054 | 0.3815 | 0.3793 | 0.4807 | 0.2474 | 0.3051 | 0.3444 |
| LogRank(Neo-2.7) | 0.8640 | 0.8635 | 0.7060 | 0.8112 | 0.7628 | 0.9143 | 0.7789 | 0.8187 |
| LRR(T5-11B/Neo-2.7) | 0.8391 | 0.8040 | 0.6596 | 0.7676 | 0.7845 | 0.8413 | 0.7067 | 0.7775 |
| NPR(T5-11B/Neo-2.7) | 0.5121 | 0.5530 | 0.4753 | 0.5135 | 0.4250 | 0.7577 | 0.5198 | 0.5675 |
| DetectGPT(T5-11B/GPT-2) | 0.4864 | 0.5698 | 0.6004 | 0.5522 | 0.2919 | 0.7035 | 0.6026 | 0.5326 |
| DetectGPT(T5-11B/Neo-2.7) | 0.5364 | 0.5172 | 0.4763 | 0.5100 | 0.3438 | 0.6894 | 0.5073 | 0.5135 |
| DetectGPT(T5-11B/GPT-J) | 0.4298 | 0.4689 | 0.4079 | 0.4355 | 0.2889 | 0.6573 | 0.4371 | 0.4611 |
| Fast(GPT-J/GPT-2) | 0.9233 | 0.9186 | **0.7727** | **0.8715** | 0.7909 | 0.9524 | **0.7697** | 0.8377 |
| Fast(GPT-J/Neo-2.7) | **0.9646** | 0.9190 | 0.7172 | 0.8669 | 0.8598 | 0.9622 | 0.7487 | 0.8569 |
| Fast(GPT-J/GPT-J) | 0.9591 | **0.9476** | 0.7063 | 0.8710 | **0.8750** | **0.9811** | 0.7428 | **0.8663** |

Table 11: Detection of ChatGPT generations under *attack*. We use the black-box settings for all zero-shot methods. We paraphrase each sentence in the ChatGPT-generated passages using T5 paraphraser for paraphrasing attack and randomly swap two adjacent words in each sentence with more than 20 words, where both attacks downgrade the coherence of the original passages.

We assess Fast-DetectGPT alongside with other zero-shot methods to evaluate their resilience in the face of adversarial attacks. Following the approach outlined in Sadasivan et al. (2023), we employed a T5-based paraphraser[4] to paraphrase text generated by ChatGPT before detection. As shown in Table 11 (Appendix F.2), all methods witnessed a decline in detection accuracy. Specifically, RoBERTa-base's AUROC decreases from 0.7474 to 0.6477, DetectGPT from 0.8342 to 0.5522, and Fast-DetectGPT from 0.9641 to 0.8715, where Fast-DetectGPT experiences the smallest relative downgrade.

However, upon detailed examination, we identify an unforeseen consequence of the paraphrasing attack: a noticeable reduction in cross-sentential coherence within passages, as an example illustrated below. This issue largely stems from the independent paraphrasing of individual sentences. We speculate that this diminished coherence is primarily responsible for the observed performance drop, given that the paraphrased text appears aberrant in its coherence relative to both machine-generated and human-authored passages.

Paraphrasing attack downgrades the coherence of the passages. For instance, consider a news report about a car crash, originally reading, "*If the car driver was hit first from behind or in front of him only on a single lap you should take control of your car. You have to be as much ahead of the car driver as possible and if you not get to your proper position with the car and the driver can't make that turn.*" The second sentence was paraphrased to, "*If you not get to your proper position with the*

---

[4]https://huggingface.co/Vamsi/T5_Paraphrase_Paws

*car and the driver can not do that turn, you need to be as much ahead of the car driver as possible.*"
When placed back in its context, we observed that the paraphrased sentence was considerably more challenging to comprehend than the original.

To test this conjecture, we execute a *decoherence attack*, wherein two adjacent words in sentences exceeding 20 words are randomly transposed. While this manipulation impacts fluency, the core meaning largely persists. As evidenced in Table 11 in Appendix, there was a comparable drop in detection accuracy, thereby empirically confirming our speculation.

## G    ADDITIONAL DISCUSSION

**Authorship of Text.** The conditional probability curvature serves as an indicator of how well a candidate passage aligns with a specific source model. When we utilize various source models, we observe varying sample discrepancies, which can aid in identifying the most suitable match among these source models. Consequently, this approach has the potential to be employed for source model identification within a set of available models.

**Watermarking.** Another line of detection methodology is watermarking that deliberately embeds information within machine-generated text to trace its origin (Jalil & Mirza, 2009; Kamaruddin et al., 2018; Abdelnabi & Fritz, 2021; Gu et al., 2022; Kirchenbauer et al., 2023). In comparison, Fast-DetectGPT relies on the innate distinction between the texts generated by humans and by machines, which may further be strengthened by explicit watermarks as additional features.

In practice, these two strategies could potentially be combined to provide a more reliable detection solution. On the one hand, watermarking can be used to authorize the content generated by a specific service. On the other hand, when the service is out of our control and we cannot enforce the watermarking or a potential attacker has a strong LLM to remove the watermarks, the watermarking approach fails in these situations but the general detector like Fast-DetectGPT can still provide a valid solution.

