# OpenReview forum: "Fast-DetectGPT: Efficient Zero-Shot Detection of Machine-Generated Text via Conditional Probability Curvature"
_ICLR.cc/2024/Conference — ICLR 2024 poster_

### Official Review · Reviewer_nx1F · 2023-10-28

**Soundness:** 3 good
**Presentation:** 3 good
**Contribution:** 2 fair
**Rating:** 8
**Confidence:** 4

**Summary:**

This paper introduces an extension of DetectGPT called Fast-DetectGPT, which modifies the curvature criterion to operate per token, using the difference between the observed and average log probabilities. This requires only a single parallel forward pass from each model.

**Strengths:**

The proposed idea is intuitive. The conditional probability function is naturally parallelized by autoregressive models and this value should naturally be close to a local maximum for a model that generated a given text. The connection to likelihood and entropy was also interesting.

The experimental results are strong and comprehensive. Fast-DetectGPT is faster than DetectGPT by over two orders of magnitude due to its parallelization and also shows performance gains across six datasets. Even in the black-box (surrogate) evaluation setting, DetectGPT achieves impressively high recall at low false positive rates. It also shows qualitatively better behavior than DetectGPT on longer passages, where quirks of T5 masking cause DetectGPT to start underperforming as sequence length increases beyond a point.

**Weaknesses:**

The end of section 2 shows that the criterion for Fast-DetectGPT can be seen as closely related to likelihood and entropy. While this connection is nice, I think the paper could be stronger if it analyzed each term in (7) in isolation to see what is most contributing to increased performance and why. Both likelihood and entropy are points of comparison in the result tables, but they do not perform as well; does their sum perform well? If not, and the denominator in (7) plays a key role, what probabilistic interpretation does that have, and what does that imply about the log_p surfaces of LLMs?

Not really a weakness and perhaps out of scope for this submission, but I'd be interested in knowing how Fast-DetectGPT would work for very long passages, given that it scales favorably with passage length.

**Questions:**

Please see sections for strengths and weaknesses.

---

> ### Author Response · Authors · 2023-11-17
>
> Thank you for your constructive feedback.
>
> **Q1**: Both likelihood and entropy are points of comparison in the result tables, but they do not perform as well; does their sum perform well? What does that imply about the log_p surfaces of LLMs?
>
> **A1**: We have revised the paper to include a segment titled “Entropy Ablation” in Appendix C on page 18 to discuss it in detail. Briefly speaking, among the total 75% relative improvement, 10% is attributed to normalization, while the remaining 65% stems from the contribution of the numerator (log-likelihood + entropy). The entropy plays a crucial role in achieving high detection accuracy in Fast-DetectGPT.
>
> The significance of entropy lies in stabilizing detection statistics by reducing variance in token-level log-likelihood across diverse contexts. The subtraction of $\log p_{\theta}(x_j|x_{<j})$ and $\tilde{\mu}_j$ results in a more stable statistic, robust to token or context fluctuations. In an experiment with ChatGPT generations for XSum, the average standard deviation reduces from 2.1893 (log-likelihood) to 1.6342 (log-likelihood + entropy), underscoring the efficacy of entropy in enhancing stability and reliability in the detection process.
>
>
> **Q2**: I'd be interested in knowing how Fast-DetectGPT would work for very long passages, given that it scales favorably with passage length.
>
> **A2**: Theoretically, the longer the passage is, the higher the detection accuracy will be, due to the statistical nature of the conditional probability curvature. However, the accuracy may not increase unlimited close to 1.0 given that the amount of increase in detection accuracy becomes marginal for longer passages. For example, when we increase the length from 120 to 150 and 180 tokens, the detection accuracy increases from 0.8864 to 0.9038 and 0.9061. The amount of increase between 0.9038 and 0.9061 is marginal compared to the increase between 0.8864 and 0.9038.

---

### Official Review · Reviewer_ubhC · 2023-10-30

**Soundness:** 3 good
**Presentation:** 2 fair
**Contribution:** 2 fair
**Rating:** 6
**Confidence:** 4

**Summary:**

This paper provides a state-of-the-art approach to zero-shot detection of LLM-generated text based on the difference of text likelihood and entropy. The paper provides extensive experiments, outperforming DetectGPT and a number of statistical baselines, as well as a supervised RoBERTa-based approach. The approach performs especially well when the scoring and target LLMs differ, e.g., when using GPT-J to detect whether an article was written by ChatGPT or GPT-4, which is a known failure mode of the existing DetectGPT approach. The paper also includes a number of experiments on different decoding strategies, reports performance across document lengths, and experiments with paraphrasing attacks.

**Strengths:**

The main strength of this work is the performance of the proposed method, which is better than much more computationally intensive zero-shot detectors such as DetectGPT. The set of ablation experiments (across languages, domains, decoding strategies, and paraphrase attacks) is also reasonably thorough, and the proposed method shows state-of-the-art performance across almost all tested conditions and datasets.

**Weaknesses:**

I find the framing of this paper and its comparison to be somewhat misleading. In particular, while the proposed method is described as a more efficient alternative to DetectGPT, its approach of computing the difference between the conditional probabilities of words and their alternatives is more similar to likelihood-based (Solaiman et al. 2019) or rank-based (GLTR; Gehrmann et al. 2019) approaches. Framing the method as a 340x speedup over DetectGPT therefore does not seem appropriate, although the method does seem to outperform existing zero-shot approaches. The sampling step in Fast-DetectGPT is also not clearly motivated and straightforwardly approximates an expected difference, so IMO the derivation could just immediately be replaced by the analytical solution.

The paper also includes supervised RoBERTa baselines from OpenAI; however, these are not state-of-the-art for supervised detection. I believe the paper would be strengthened by comparison to state-of-the-art supervised methods, such as Ghostbuster (Verma et al. 2023) or GPTZero (commercial model), especially given the claims in Section 5 that supervised methods have limited generalization capabilities in LLM-generated text detection. Because the primary purpose of the paper is to evaluate and compare zero-shot methods, however, this does not affect my score or recommendation for the paper.

Minor notes:
- The paper mentions both Rank and LogRank baselines in Section 3.1 but only provides LogRank in tables

**Questions:**

- Did you experiment with computing the difference between the probability of the top-ranked word according to an LM scorer and the observed word? I expect this should be closely correlated with the metric proposed in this paper, and is also a slightly more informative alternative to the Rank model.

---

> ### Author Response · Authors · 2023-11-17
>
> Thank you for your constructive feedback.
>
> **Q1**: I believe the paper would be strengthened by comparison to state-of-the-art supervised methods, such as Ghostbuster (Verma et al. 2023) or GPTZero (commercial model).
>
> **A1**: We have revised the paper to include GPTZero as a baseline as Table 3 on page 7 and Table 7 on page 17 show. In summary, Fast-DetectGPT obtains higher average AUROCs than GPTZero. Specifically, on GPT-3 generations, GPTZero demonstrates poor detection accuracy on all three datasets, suggesting that GPTZero may not be trained on GPT-3 generations. On Chat-GPT and GPT-4 generations, GPTZero performs better than Fast-DetectGPT on news (XSum) but worse on stories (Writing) and technical articles (PubMed), which suggests that GPTZero may mainly be trained on news generations.
>
> **Q2**: Did you experiment with computing the difference between the probability of the top-ranked word according to an LM scorer and the observed word?
>
> **A2**: That is an interesting idea. We just experimented with it on GPT-4 generations and obtained an average AUROC of 0.8542, which is better than the likelihood (0.8212) and the log-rank (0.8088) but worse than Fast-Detect (0.9061). The results suggest that the probability difference is a better criterion than previous simple baselines.
>
> **Q3**: The sampling step in Fast-DetectGPT is also not clearly motivated and straightforwardly approximates an expected difference, so IMO the derivation could just immediately be replaced by the analytical solution.
>
> **A3**: We have revised the paper to introduce an individual segment titled “Conditional Independent Sampling” to discuss it in detail.
>
> The sampling process plays a pivotal role in guiding us toward the solution. To discern whether a token within a given context is machine-generated or human-authored, it is essential to compare it against a range of alternative tokens in the same context. By sampling a substantial number of alternatives (say 10,000), we can effectively map out the distribution of their $\log p_{\theta}(\tilde{x}_ j|x_{<j})$ values. Placing the $\log p_{\theta}(x_ j|x_{<j})$ value of the passage token within this distribution provides a clear view of its relative position, enabling us to ascertain whether it is an outlier or a more typical selection. This fundamental insight forms the core rationale behind the development of Fast-DetectGPT.
>
> The analytical solution is computationally more efficient but intuitively harder to understand, which we did not anticipate at the early stage of the research. Furthermore, when we use a single model for both sampling and scoring, the analytical solution shows a close connection to the Likelihood and Entropy baselines as we described on page 5. This connection seems so simple as the numerator in Eq. 7 is just the sum of Likelihood and Entropy. However, the sampling intuition plays a key role in finding this connection, where the Likelihood and Entropy have been used since year 2008.
>
> **Q4**: The paper mentions both Rank and LogRank baselines in Section 3.1 but only provides LogRank in tables.
>
> **A4**: We have revised the paper to remove the Rank baseline. Since Rank generally performs worse than LogRank, we remove it from the table to save space.

---

### Official Review · Reviewer_aeWx · 2023-11-01

**Soundness:** 3 good
**Presentation:** 4 excellent
**Contribution:** 4 excellent
**Rating:** 8
**Confidence:** 4

**Summary:**

This paper proposes a new method for detecting LLM generated text that offers not only substantial performance benefits over DetectGPT but is also much less compute intensive. This is underpinned by a hypothesis that context matters in determining the differences between human and machine generated output. Their method accordingly uses a new criteria, the conditional probability curvature, which they find is more positive for LLM output than human. They perform experiments on a variety of datasets, and analyze robustness with respect to multiple text attributes.

**Strengths:**

- The proposed method is well motivated and described, and follows naturally from existing work
- The results are strong both from a performance and efficiency standpoint, compared to DetectGPT
- There is meaningful analysis with respect to attributes like passage length, paraphrasing, decoding strategies, etc.

**Weaknesses:**

- The discussion of prior work with respect to alternate detection strategies such as watermarking is shallow. The Kirchenbauer et al. 2023 paper is for example not cited. While this paper takes an orthogonal approach, it would be good to see some motivation or discussion around the tradeoffs of those strategies.
- The discussion of ethical considerations and broader impacts is lacking. Liang et al. 2023 has shown that LLM detection systems tend to exhibit higher false positive rates for non-native speakers. While this doesn’t invalidate the usefulness of this work, at the least it is worth engaging with that literature and acknowledging the potential problems at play with this task. At best there could be experiments on the relative performance of this system on text written by different demographics as compared to prior work. Granted there is some analysis of performance on languages besides English but this is also relatively shallow.

**Questions:**

Have you investigated the effects of varying the temperature setting or the value of k for Top-k?

---

> ### Author Response · Authors · 2023-11-17
>
> Thank you for your constructive feedback.
>
> **Q1**: Have you investigated the effects of varying the temperature setting or the value of k for Top-k?
>
> **A1**: We have revised the paper to include our experiments on a temperature of 0.6, top-k of 30, and top-p of 0.90 in Table 9 on page 19 and a detailed description in Appendix D.1 on page 20. Briefly speaking, reducing the values of T, k, and p enhances the determinism of generated samples, facilitating easier detection and resulting in higher AUROCs.
>
>
> **Q2**: The discussion of prior work with respect to alternate detection strategies such as watermarking is shallow. The Kirchenbauer et al. 2023 paper is for example not cited. It would be good to see some motivation or discussion around the tradeoffs of those strategies.
>
> **A2**: We have revised the paper to add a deeper discussion about watermarking in Appendix E on page 21, where the content is as follows.
>
> Another line of detection methodology is watermarking that deliberately embeds information within machine-generated text to trace its origin \citep{jalil2009review, kamaruddin2018review, abdelnabi2021adversarial, gu2022watermarking, kirchenbauer2023watermark}. In comparison, Fast-DetectGPT relies on the innate distinction between the texts generated by humans and by machines, which may further be strengthened by explicit watermarks as additional features.
>
> In practice, these two strategies could potentially be combined to provide a more reliable detection solution. On the one hand, watermarking can be used to authorize the content generated by a specific service. On the other hand, when the service is out of our control and we cannot enforce the watermarking or a potential attacker has a strong LLM to remove the watermarks, the watermarking approach fails in these situations but the general detector like Fast-DetectGPT can still provide a valid solution.
>
> **Q3**: The discussion of ethical considerations and broader impacts is lacking.
>
> **A3**: We have revised the paper to add a segment “Ethical Considerations and Broader Impact” in Appendix E on page 21 with the following content.
>
> Fast-DetectGPT, serving as a highly efficient detector for machine-generated text, holds promise in enhancing the integrity of AI systems by combating issues like fake news, disinformation, and academic plagiarism. However, akin to other methods reliant on Large Language Models (LLMs), it is susceptible to inheriting biases present in the training data. Notably, as emphasized by \cite{liang2023gpt}, LLM-based detection systems may exhibit an elevated false-positive rate when confronted with text from non-native English speakers. Given the widespread and diverse utilization of such technologies, this presents a notable concern.
>
> An immediate suggestion is to substitute the underlying LLMs in Fast-DetectGPT with alternative models trained on more varied and representative corpora. Additionally, we advocate for community involvement in the ongoing efforts to develop more inclusive LLMs, a development that would benefit not only Fast-DetectGPT but also similar systems at large.

---

### Official Review · Reviewer_7WUs · 2023-11-01

**Soundness:** 3 good
**Presentation:** 2 fair
**Contribution:** 3 good
**Rating:** 6
**Confidence:** 3

**Summary:**

This paper present an extension for DetectGPT, improving its efficiency and effectiveness. Relying on LLM's output probability, the model can threshold and perform zero-shot detection. Given a sentence, the model will first autoregressively predict x' from the input, and then use the original input x as input to a LLM but calculate the probability to predict x'. The modification is simple, and effective, which intuitively makes sense.

**Strengths:**

1. Improved results over DetectGPT for 3 points, with also faster speed.

2. The paper also showed results on detect GPT-4 results.

3. Analytical solution presented to avoid sampling approximation.

4. Ablation study on different lengths, decoding strategies, paraphrasing has been shown.

**Weaknesses:**

1. Presentation should be made clear. In the intro, paragraph 4 talked about the algorithm, yet it is unclear what does \tilt mean, what does <j means, also, the insight on why conditional probability is better is missing here, especially given that this is an extension of DetectGPT.

2. Is there results for speed comparison?

**Questions:**

Can you elaborate how \tilt {x} is generated? The reviewer is still confused.

Where does the acceleration come from? DetectGPT samples 100 pertrubations, how could this method accelerate 340 times? How many sampling does this needs?

---

> ### Author Response · Authors · 2023-11-17
>
> Thank you for your constructive feedback.
>
> **Q1**: In the intro, paragraph 4 talked about the algorithm, yet it is unclear what \tilde mean, what does <j means. The insight on why conditional probability is better is missing here. Can you elaborate on how \tilde{x} is generated? How many sampling does this needs? Where does the acceleration come from?
>
> **A1**: We have made revisions to enhance the clarity of the paper. Please refer to the improved paragraph 4 in the introduction and the newly added section titled "Conditional Independent Sampling" on page 4 for detailed explanations.
>
> Briefly speaking, $\tilde{x}$ is generated by sampling each token $\tilde{x}_ j$ independently from the conditional probability $p(\tilde{x}_ j|x_{<j})$, where the passage $x$ is fixed. The independence of $\tilde{x}_j$ given the passage $x$ is the key to the high efficiency of Fast-DetectGPT. Specifically, Fast-DetectGPT requires only a single forward pass to sample and evaluate 10,000 samples (our default setting), in contrast to DetectGPT, which typically necessitates 100 forward passes for a mere 100 perturbations. This results in a significant reduction in computation time and resources.
>
> **Q2**: Is there results for speed comparison?
>
> **A2**: We compared the speedup in Table 1 on page 1 and we have revised the paper to include a new segment titled “Inference Speedup” on page 7, providing a detailed description.
>
> Briefly speaking, for the detection of XSum generations produced by the 5-model. DetectGPT total costs (14598s+13819s+14383s+15941s+20372s) = 79113s, which is about 22 hours. In contrast, Fast-DetectGPT total costs (34s+31s+34s+49s+85s) = 233s, which is about 4 minutes. The speedup is calculated by 79113s / 233s, which is around 340x.

---

### Official Review · Reviewer_RgLj · 2023-11-04

**Soundness:** 2 fair
**Presentation:** 2 fair
**Contribution:** 1 poor
**Rating:** 6
**Confidence:** 5

**Summary:**

This paper improves the previous zero-shot method for detecting machine-generated text, DetectGPT, by replacing the perturbations as sampling using the same source model. Through the conditional probability curve, the author proves the effectiveness of this method. However, some experimental details are missing. More importantly, it did not mention another zero-shot work [1] released 5 months ago, which is the first to propose using a conditional probability curve for detection. Considering the similarity with the previous work [1], I would like to question the novelty of this paper since the long 5-month period clearly shows they are not concurrent work.

**Strengths:**

Strength:
The experiments over diverse datasets and models validate its effectiveness.
The author considers both open-sourced and closed-source models for the detection. Thus, the results can be easily reproduced.
The ablation study is enough to support its claim regarding parameter sensitivity, attacks, etc.
The paper is well-written and easy to follow. The tables and figures are arranged properly.

Missing reference:
The following zero-shot method is missing either in the related work or in the baselines.
[1] Yang X, Cheng W, Petzold L, Wang WY, Chen H. DNA-GPT: Divergent N-Gram Analysis for Training-Free Detection of GPT-Generated Text. arXiv preprint arXiv:2305.17359. 2023 May 27.

**Weaknesses:**

Weakness:
1. The novelty is limited. The conditional probability curve has already been used by another zero-shot detector released 5 months ago [1]. However, the author neither cites this previous work nor discusses its differences. Considering the reference [1] work was released 5 months ago, I will not consider them as concurrent work.
2. It is not clear how the sampling process works. Give a passage x, how do you sample the alternative x’ ? Throughout the paper, I did not find any explanation for this.
3. How would the number of resampled instances influence the result? I did not find any result for this.
4. What is your default setting for the number of resampled instances for all the experiments? There is no clarification at all.
5. How do you compare the speedup of your result over DetectGPT? Since the setting of your number of samples is unclear, I am not sure how did you compare it.


After rebuttal: Thanks for the clarification. The authors addressed most of my concerns. I would like to raise my score.

**Questions:**

Questions:
The number of relative improvements is confusing. For example, in Table 1, why is the relative improvement 74.7%? In my understanding, (0.9887−0.9554)/0.9554*100%=3.48%. I do not understand why you report 74.7%.
See more in Weakness.

---

> ### Author Response · Authors · 2023-11-17
> **Author response (Part1)**
>
> Thank you for your feedback.
>
> **Q1**: Missing reference: The following zero-shot method is missing either in the related work or in the baselines.
>
> **A1**: We have revised the paper to incorporate DNA-GPT as a baseline. Please refer to Tables 2, 3, 5, 7, and the relevant sections on related work for details.
>
> **Q2**: The novelty is limited. The conditional probability curve has already been used by another zero-shot detector released 5 months ago [1].
>
> **A2**: We **strongly disagree** with this statement given the following evidence.
>
> First, **neither the term “conditional probability curvature“ nor terms with similar meanings are found in the paper** including both arXiv versions published on 5/27/2023 (https://arxiv.org/abs/2305.17359v1) and on 10/4/2023 (https://arxiv.org/abs/2305.17359v2), where the second version is even after our submission to ICLR 2024 on 9/28/2023.
>
> Second, **the basic idea of DNA-GPT is fundamentally different from that of Fast-DetectGPT**. DNA-GPT generates various completions given a truncated prefix of a candidate passage, where the completions are generated autoregressively (requiring a decoding process for each completion). In contrast, Fast-DetectGPT does token-level conditionally independent sampling, where all samplings are done in the predictive distribution (requiring only one forward pass instead of any decoding process).
>
> Third, **DNA-GPT costs 80x more inference time than Fast-DetectGPT**. Experiments on XSum generations produced by the 5-model show that DNA-GPT takes 19289s (about 5 and half hours) to run across the five models, while Fast-DetectGPT only takes 233s (about 4 minutes). DNA-GPT costs 82.8x the inference time of Fast-DetectGPT.
>
> Last, **DNA-GPT has much worse detection accuracy than Fast-DetectGPT** on both 5-model generations and ChatGPT/GPT-4 generations.  As Tables 2, 3, 5, and 7 in the revised version show, DNA-GPT performs even worse than DetectGPT on 5-model generations. Although its detection accuracy on ChatGPT and GPT-4 generations is higher than DetectGPT but is still significantly lower than Fast-DetectGPT (0.8836 and 0.7648 for DNA-GPT vs. 0.9615 and 0.9061 for Fast-DetectGPT on ChatGPT and GPT-4, respectively).
>
> **Q3**: It is not clear how the sampling process works. Give a passage x, how do you sample the alternative x’?
>
> **A3**: We mentioned it in various places and in various forms, for example, “the conditional probabilities of alternative tokens p(\tilde{x}_j|x_{<j})” and “our approach begins by **sampling alternative word choices at each token**” on page 2, the definition of conditional probability function “p(\tilde{x}|x)” and “conditional sampling” part of Algorithm 1 on page 4, and the analytical expression of the sample mean in Eq. 5 on page 5.
>
> The sampling of alternative \tilde{x} is conditionally independent given a passage x that each token \tilde{x}_j is sampled from the conditional distribution of p(\tilde{x}_j|x_{<j}) independently. We have revised the paper to introduce an additional segment titled “Conditionally Independent Sampling” to illustrate the sampling process in detail.
>
> **Q4**: How would the number of resampled instances influence the result?
>
> **A4**: As we mentioned in “the analytical solution achieves a detection accuracy **almost identical** to the sampling approximation with 10,000 samples” on page 5, sampling with the number of resampled instances from 10,000 to 50,000 produces NO obvious differences in the AUROC, and also NO obvious difference in the time cost.
>
> **Q5**: What is your default setting for the number of resampled instances for all the experiments?
>
> **A5**: We by default use 10,000 samples for the experiments as we mentioned in “the analytical solution achieves a detection accuracy almost identical to the sampling approximation with **10,000 samples**” on page 5.
>
> **Q6**: How do you compare the speedup of your result over DetectGPT?
>
> **A6**: We mentioned it in the caption of Table 1 “Speedup assessments were conducted **using the XSum** news dataset, with computations **on a Tesla A100 GPU**”, and we further revised the paper to include a new segment “Inference Speedup” on page 7 to discuss it in detail.
>
> Specifically, we use 10,000 samples for Fast-DetectGPT and the default setting of 100 perturbations for DetectGPT. The speedup is evaluated on XSum generations from the 5 models. DetectGPT exhibited substantial computational demands, with total processing times across five runs summing up to 79,113 seconds (14598s+13819s+14383s+15941s+20372s, approximately 22 hours). Conversely, Fast-DetectGPT demonstrated exceptional efficiency with total times of only 233 seconds (34s+31s+34s+49s+85s, about 4 minutes). The resulting speedup factor is approximately 340x, calculated by dividing the total time for DetectGPT by that for Fast-DetectGPT (79,113s / 233s ≈ 340).

---

> > ### Comment · Reviewer_RgLj · 2023-11-22
> > **Thanks for the clarification**
> >
> > Thanks for the clarification. I have modified my score.

---

> ### Author Response · Authors · 2023-11-17
> **Author response (Part 2)**
>
> **Q7**: The number of relative improvements is confusing.
>
> **A7**: We have revised the paper to clarify it as Table 1 on page 1 shows. The relative improvements are calculated by (new – old) / (1.0 – old), where the denominator (1.0 – old) represents the maximum possible improvement from the old AUROC. The metric shows how much improvement has been made relative to the maximum possible improvement, which is especially useful when the values of new and old are already high (close to 1). For example, intuitively, an improvement from 0.9554 to 0.9887 looks so different from an improvement from 0.7225 to 0.9338. However, when we compare their relative improvements, we can see that the former 74.7% is close to the latter 76.1%, which reveals the inner consistency between the two experiments.

---

### Public Comment · ~Eric_Mitchell1 · 2023-11-22
**External opinion: Clearly a good paper and should be accepted**

I have no affiliation with the authors of this paper (in fact I do not even know who they are or what organization they are from), but I am very familiar with the area (I am the lead author of the original DetectGPT paper this work builds on).

I feel the need to weigh in here only in light of the (very surprising) low score this paper received.

The insight the authors contribute is clever, novel, and makes practically meaningful improvements on existing detectors along all three dimensions of efficiency, simplicity of implementation, and detection quality; this combination of contributions is rare. These factors were clear to 4 out of 5 reviewers, who all felt the paper should be accepted; they particularly appreciated the elegance of the proposed solution and the comprehensiveness of the experiments.

Having read the full paper myself, I agree with 4/5 reviewers; specifically, it is my view that **this is clearly a good paper and without doubt it should be accepted to ICLR.**

---

> ### Author Response · Authors · 2023-11-23
>
> Dear Eric Mitchell,
>
> We want to express our deepest gratitude for taking the time to provide such a thoughtful and supportive public comment on our ICLR paper. Your recognition, especially considering your expertise as the lead author of the original DetectGPT paper, means a great deal to us. We are truly honored that someone of your caliber acknowledges the cleverness and novelty of our contributions, as well as the improvements we've made in efficiency, simplicity of implementation, and detection quality.
>
> We are encouraged by your positive evaluation, and we look forward to the opportunity to discuss our findings further. Your insights and perspective are invaluable to us, and we are committed to continuing our efforts to contribute meaningfully to the field.
>
> Once again, thank you for your generous support. We are truly grateful.

---

### Meta-Review · Area_Chair_Dvua · 2023-12-13

**Metareview:**

This paper presents a new method for zero-shot machine-generated text detection based on DetectGPT. In contrast with DetectGPT, the proposed method generates simple conditional perturbations of the input by independently resampling each word conditioned on its left context. The resulting detector outperforms DetectGPT as well as other state-of-the-art zero-shot detectors, while being substantially faster than DetectGPT. Reviews are generally in favor of acceptance, nearly universally praising the method's performance and simplicity. Weaknesses brought up by reviewers included missing comparisons with non-zero-shot baselines for added context (added in rebuttal), issues with clarity of presentation (addressed in rebuttal), and the observation that relative to baselines other than DetectGPT, speeds-ups are less pronounced.

**Justification For Why Not Higher Score:**

While the method proposed in this paper is practical and useful, it is not fundamentally different from methods that came before it -- it represents a targeted and simple extension with practical benefits.

**Justification For Why Not Lower Score:**

The propose method is simple and effective -- it achieves state-of-the-art zero-shot detection while being computationally lightweight.

---

### Decision · Program_Chairs · 2024-01-16

Accept (poster)